# MINDFeed: Mutual Information-Guided Single-Network Consistency Learning for Semi-Supervised 3D Medical Image Segmentation

## Abstract

Medical image segmentation models based on deep learning require dense voxel-level annotations, which are costly to obtain for 3D medical imaging tasks. To address this limitation, we propose MINDFeed (Mutual Information per Decoder as Feedback), a semi-supervised training pipeline for 3D medical image segmentation. MINDFeed estimates predictive uncertainty via mutual information across stochastic forward passes and uses this signal to adaptively modulate decoder representations as a feedback gate, encouraging consistency in reliable regions while suppressing ambiguous responses. Unlike many prior approaches, MINDFeed does not rely on student–teacher architectures, exponential moving averages, or multiple model instances, thereby maintaining architectural simplicity and training efficiency. We conduct extensive experiments on CT and MRI datasets, covering binary and multi-class segmentation tasks with both single- and multi-modal inputs, and demonstrate that MINDFeed consistently outperforms recent state-of-the-art semi-supervised methods. In addition to improved segmentation performance, MINDFeed exhibits reduced variability among test samples, highlighting its robustness under limited annotation settings. We provide our code at https://anonymous.4open.science/r/MINDFeed-A306.

## 1 Introduction

Medical image segmentation is a fundamental task in computer-aided diagnosis and treatment planning, where accurate delineation of anatomical structures and pathological regions is essential. In many clinical applications, medical data are acquired as volumetric scans, necessitating robust 3D segmentation methods. Compared to 2D settings, 3D medical image segmentation introduces additional challenges due to high dimensionality, increased computational cost, and greater annotation complexity.

Recent advances in deep learning and computer vision have led to the development of numerous methods for medical image segmentation. These methods come in the form of training pipelines and model architectures. Several architectures have achieved state-of-the-art results in medical image segmentation using fully supervised pipelines. However, in Semi-Supervised Learning scenarios where labeled data is limited, both the choice of the model and the training pipeline become critical factors in achieving optimal performance.

Numerous Semi-Supervised methods have been developed over the years since the introduction of the Mean Teacher approach (Tarvainen & Valpola, 2017), introducing and incorporating new ideas such as adversarial training (Zhang et al., 2017), pseudo-labeling (Sohn et al., 2020), uncertainty estimation (Yu et al., 2019; Luo et al., 2022b; Grandvalet & Bengio, 2004; Peng et al., 2021; 2020), cross-model consistency training mechanisms (Luo et al., 2022a; Wang et al., 2023), and many others. Existing methods often introduce additional architectural complexity through multiple model instances or auxiliary teacher networks, and typically exploit uncertainty only as a training-time heuristic, such as loss weighting or a decision-making criterion. Evaluation of semi-supervised segmentation methods is often restricted to simplified settings such as single-modality or binary segmentation tasks. 3D medical image segmentation often requires the effective use of multiple modalities and the segmentation of different regions within the same volume.

The main contributions of this paper are as follows.

1. We introduce an uncertainty-aware learning paradigm in which voxel-level epistemic uncertainty is incorporated directly into intermediate feature representations as a feedback signal. Unlike prior approaches that use uncertainty primarily as modifications to losses or decision-making, our method allows uncertainty to influence the forward feature transformation itself.

2. Building on this paradigm, we propose MINDFeed, a single-network semi-supervised framework that leverages decoder-level mutual information as a feedback gating mechanism to enforce internal consistency during training, without relying on teacher–student models, exponential moving averages, or multiple network instances.

3. We extensively evaluate the proposed method on four public 3D medical image segmentation benchmarks, including BraTS 2019, BraTS-GLI 2024, LA 2018, and NIH Pancreas-CT. Experimental results demonstrate that MINDFeed achieves consistent performance improvements over recent semi-supervised approaches across binary and multi-label segmentation tasks, as well as single- and multi-modal inputs, while also reducing variability among test cases.

The rest of this paper is organized as follows: Section 2 provides a brief literature review. Section 3 details the proposed approach, introducing its main components and the entire training pipeline. Section 4 presents the experimental setup, obtained results, and ablation studies. Section 5 provides an analysis of the findings, and Section 6 concludes the paper.

## 2 Related Work

### 2.1 Semi-Supervised Learning for Medical Image Segmentation

Several semi-supervised learning (SSL) approaches have been proposed for medical image segmentation. While some of these methods were originally developed for classification tasks, they have been successfully adapted to segmentation, demonstrating improved performance over fully supervised approaches when labeled data is limited. The Mean Teacher (Tarvainen & Valpola, 2017) approach introduced the use of a student–teacher framework with consistency regularization between predictions on augmented inputs, providing a foundation for many subsequent methods. Adversarial SSL approaches began with Deep Adversarial Networks (DANs) (Zhang et al., 2017), where two models are trained adversarially to enhance each other's performance. Pseudo-labeling strategies have also been widely used: for instance, FixMatch (Sohn et al., 2020) combines weak and strong augmentations with confidence thresholding to produce more reliable pseudo-labels, while Cross Consistency Training (CCT) (Ouali et al., 2020) enforces consistency across multiple augmented views to improve robustness. Variations in architectures that use multiple model instances have also been explored, such as cross-teaching between CNNs and Transformers (Luo et al., 2022a) and Mutual Correction Framework (MCF) (Wang et al., 2023). Other approaches, such as Dual Task Consistency (DTC) (Luo et al., 2021) and Cross Pseudo Supervision (CPS) (Chen et al., 2021), employ alternative strategies. DTC integrates the signed distance function into its loss function with a single model instance, while CPS uses a linear combination of losses from multiple model instances (2 is usually used).

While these methods have demonstrated improved performance with limited labeled data, some, such as the Mean Teacher variants, require maintaining multiple model instances (student and teacher), increasing computational cost. Approaches relying on pseudo-labeling, including FixMatch and Cross Consistency Training, are often limited by the quality of their pseudo-labels, which can reduce reliability and robustness. Moreover, none of the methods discussed in this section explicitly models voxel-level uncertainty, adopting a deterministic or frequentist approach rather than a probabilistic one, which has been reported to be beneficial in ambiguous or boundary regions.

In the context of medical image segmentation, semi-supervised methods face the added complexity of multi-label segmentation, where multiple subregions (such as brain tumors) have to be identified within the same volume. Many existing semi-supervised methods are primarily evaluated on binary segmentation tasks. Unlike binary segmentation tasks, multilabel segmentation requires distinguishing between regions with highly

Table 1: Summary of selected semi-supervised segmentation methods and uncertainty usage.

| Method | Single / Multiple Model Instances | Uncertainty | Uncertainty Integration |
|---|---|---|---|
| Mean Teacher (NIPS 2017) | Multiple (Same) | No | — |
| UA-MT (MICCAI 2019) | Multiple (Same) | Yes (Predictive Entropy) | Loss Modification |
| FixMatch (NeurIPS 2020) | Single | No | — |
| CCT (CVPR 2020) | Single | No | — |
| Peng et al. (MELBA 2020) (Peng et al., 2021) | Single | Yes (Mutual Information) | Loss Modification |
| CPS (CVPR 2021) | Multiple (Same/Different) | No | — |
| DTC (AAAI 2021) | Single | No | — |
| Cross-Teaching CNN+ViT (MIDL 2022) | Multiple (Different) | No | — |
| URPC (MedIA 2022) | Single | Yes (KL Divergence) | Loss Modification |
| MCF (CVPR 2023) | Multiple (Different) | No | — |
| Co-BioNet (Nat. Mach 2023) (Peiris et al., 2023) | Multiple (Different) | Yes (Adversarial Confidence) | Loss Modification |
| AC-MT (MedIA 2023) | Multiple (Same) | Yes (Label Noise Identification) | Loss Modification |
| AD-MT (ECCV 2024) | Multiple (Same) | Yes (Shannon Entropy) | Conflict Resolution |
| SGRS-Net (MICCAI 2025) | Multiple (Same) | Yes (Shannon Entropy) | Region Filtering / Masking |
| **Ours** | **Single** | **Yes (Mutual Information)** | **Feedback Gating** |

In column two, if multiple model instances are restricted to the same architecture, "Same" is indicated in parentheses; likewise, "Different" and "Same/Different" denote their respective cases.

overlapping intensity distributions and ambiguous boundaries that are difficult to separate. Moreover, the presence of severe class imbalance complicates training under limited labeled data.

## 2.2 Uncertainty Estimation in Deep Learning Models

Modeling uncertainty is crucial in medical image segmentation because deterministic predictions do not provide a measure of confidence, which can be misleading in ambiguous regions such as tumor boundaries or areas affected by noise and artifacts. By estimating uncertainty, models can identify regions where predictions are less reliable, guide more robust learning, and allow safer clinical decision-making.

Several methods have been proposed to estimate uncertainty in deep learning models (Gal, 2016; Kendall & Gal, 2017; Pocevičiūtė et al., 2022; Mukhoti et al., 2023). Common approaches include Predictive Entropy, Mutual Information, and the use of Kullback–Leibler (KL) divergence. Uncertainty is broadly categorized into epistemic and aleatoric types: epistemic uncertainty captures the model's inherent lack of knowledge and can be reduced with additional training data, while aleatoric uncertainty arises from noise or inherent ambiguity in the data and cannot be reduced through more data. While the effectiveness of uncertainty estimation methods is often task-dependent (Wimmer et al., 2023), Mutual Information (MI) and Predictive Entropy (PE) differ fundamentally in what they capture. MI quantifies epistemic uncertainty, whereas PE reflects the total (epistemic and aleatoric) uncertainty (Kendall & Gal, 2017). We hypothesize that MI provides a more accurate measure of model-intrinsic uncertainty in the context of semi-supervised 3D medical image segmentation and serves as a more informative feedback signal to the model during training. Several studies (Pocevičiūtė et al., 2022; Cangalovic et al., 2023; Kwon et al., 2020) have also reported that MI can outperform PE under specific conditions.

### 2.3    Using Uncertainty in Semi-Supervised Learning

Several approaches have been developed to estimate and include uncertainty during training, particularly in Semi-Supervised Learning. Uncertainty Aware Mean Teacher (UA-MT) (Yu et al., 2019), and Uncertainty Rectified Pyramid Consistency (URPC) (Luo et al., 2022b) are two such semi-supervised methods that estimate uncertainty using different techniques, namely Predictive Entropy and KL divergence, respectively. Other approaches leverage information-theoretic concepts, such as using Entropy (Grandvalet & Bengio, 2004) or Mutual Information (Peng et al., 2020) as a regularization term or maximizing Mutual Information between clusters of latent representations (Peng et al., 2021). Another approach, Abiguity-Consensus Mean Teacher (AC-MT) (Xu et al., 2023), uses uncertainty as an identification of ambiguity, which is incorporated as a mask in consistency regularization. More recent methods, such as Alternate Diverse Teaching (AD-MT) (Zhao et al., 2024) and Synergy-Guided Regional Supervision (SGRS-Net) (Wang et al., 2025), use Shannon Entropy (entropy of a single softmax output) for resolving conflicts between teacher models and for unreliable region filtering/masking, respectively. Most methods incorporate uncertainty into the learning process only through modified loss functions rather than as direct inputs to the model. Providing the model with voxel-wise uncertainty directly as an input can allow it to adaptively focus on ambiguous regions, improving robustness and segmentation accuracy.

We provide a summary of the comparison between certain Semi-Supervised approaches and our proposed approach in Table 1.

## 3    Methods

This section introduces the principal components of the proposed method and explains how they are incorporated into the training pipeline. The pipeline is formulated to maintain architectural flexibility, allowing its application to any segmentation model that employs multiple decoder stages, such as the VNet (Milletari et al., 2016) and 3D UNet (Çiçek et al., 2016).

### 3.1    Multi-Scale Decoder Supervision

To improve the use of the small amount of labeled samples available, we use deep supervision by adding auxiliary segmentation heads to each decoder (Luo et al., 2022b), which are used to extract the segmentation maps from each decoder block of the same size as the ground truth. This is done by adding an extra $1 \times 1 \times 1$ convolution layer to each decoder output, followed by an upsampling operation, to match the dimensions of the desired segmentation maps. The array of these $N_{decoders}$ logits is denoted by $\hat{\mathcal{D}} = \{\hat{\mathcal{D}}_1, \ldots, \hat{\mathcal{D}}_{N_{decoders}}\}$ throughout the rest of this paper.

### 3.2    Uncertainty Using Mutual Information

Mathematically, Mutual Information quantifies how much knowing one random variable reduces uncertainty about another, and is computed according to equation 1 (Shannon, 1948) given two random variables $X$ and $Y$.

$$\mathcal{I}(X;Y) = \mathcal{H}(X) - \mathcal{H}(X \mid Y) \tag{1}$$

where $\mathcal{H}(X)$ is the entropy of the random variable, $X$. In our case, we treat the model parameters $\theta$ as a random variable. This is possible since we use dropout variational inference as a Bayesian approximation (Gal & Ghahramani, 2016). Since dropout variational inference approximates sampling from the posterior distribution $p(\theta|D)$, repeated stochastic forward passes produce multiple predictive distributions $p(y|x, \theta)$ corresponding to different sampled $\theta$. This gives us the true mutual information in equation 2 (Gal, 2016). Notice how the first term in this equation is actually Predictive Entropy, which is the total uncertainty, while the second term represents the expectation of each stochastic step's entropy, which is the aleatoric uncertainty. Their difference gives us an estimate of the epistemic uncertainty.

$$\mathcal{I}(\theta;\, p(y \mid x, \theta)) = \mathcal{H}\left[\mathbb{E}_{\theta \sim p(\theta|D)}\, p(y \mid x, \theta)\right] - \mathbb{E}_{\theta \sim p(\theta|D)}\left[\mathcal{H}\left(p(y \mid x, \theta)\right)\right] \tag{2}$$

For each decoder $i$, by performing a Monte-Carlo Dropout operation with T steps on a decoder $i$ and applying a softmax to the logits at each of the T steps (obtaining a list of $T$ predictions, $MC_i$), we can approximate this using equation 3 (Gal, 2016).

$$MI_i \approx \mathcal{H}\left(\frac{1}{T}\sum_{j=1}^{T} MC_i[\, j\, ]\right) - \frac{1}{T}\sum_{j=1}^{T}\mathcal{H}(MC_i[\, j\, ]) \ \ \forall\, i \in [1, N_{decoders}] \tag{3}$$

$$\mathcal{H}(\mathbf{p}) = -\sum_{c=1}^{C} p_c \log p_c \tag{4}$$

where $\mathcal{H}(\mathbf{p})$ is the entropy of a predicted probability distribution $\mathbf{p} \in \mathbb{R}^C$, where $C$ is the number of classes.

A key novelty in our approach is the use of Mutual Information maps computed from weakly augmented inputs to guide the segmentation of strongly augmented inputs via the feedback gating mechanism. Although Mutual Information primarily captures epistemic uncertainty, it remains sensitive to input perturbations induced by data augmentation. This property is particularly relevant in our weak–strong augmentation setting (see Section 4.3), where MI maps computed from weakly augmented inputs provide stable uncertainty estimates that meaningfully differ from those obtained under strong perturbations.

### 3.3 Feedback Gating Mechanism

We introduce the use of the complement of computed Mutual Information maps as a gate on either the incoming encoder feature (via skip-connection), or the decoder output just before input to the auxiliary segmentation head used for deep supervision, for each decoder block. The choice of encoder or decoder feature gating is largely dependent on the architecture, specifically the handling of encoder and decoder fusion. A study on the determination of this choice is provided in Section 4.6.4. For the rest of the paper, unless explicitly mentioned, we use encoder gating for the VNet and decoder gating for the 3D UNet.

Regardless of whether encoder features or decoder features are used, the same gating operation is performed. Due to the shape mismatch between the gate and feature, we do a trilinear interpolation of the Mutual Information maps to the desired size, and take its complement as described in equation 5, where $f_i$ denotes the feature to be gated at decoder block $i$, and $\odot$ is the Hadamard Product. $\alpha$ is a coefficient whose value is determined based on equation 6. Since Mutual Information lies within the range $[0, \log C]$, and we use unity to take the complement, we normalize the MI maps whenever $\log C$ exceeds unity i.e., whenever $C \geq 4$, since log is the natural logarithm.

$$f_i = f_i \ \odot \ (1 - \text{interpolate}(\alpha \cdot MI_i)) \ \ \forall\, i \in [1, N_{decoders}] \tag{5}$$

$$\alpha = \frac{1}{\max(1, \log C)} \tag{6}$$

This gating mechanism softly modulates decoder features by attenuating highly uncertain responses while preserving the overall semantic structure of the feature representations, as opposed to hard gating or masking. We show this visually in Figure 1, by plotting an intermediate slice of the absolute difference between the original feature and gated feature. Mathematically, if $g_i = 1 - u_i$ represents the gate, where $u_i$ is the interpolated scaled MI map at decoder block $i$, then the difference between the original and gated feature can be expressed as shown in equation 7.

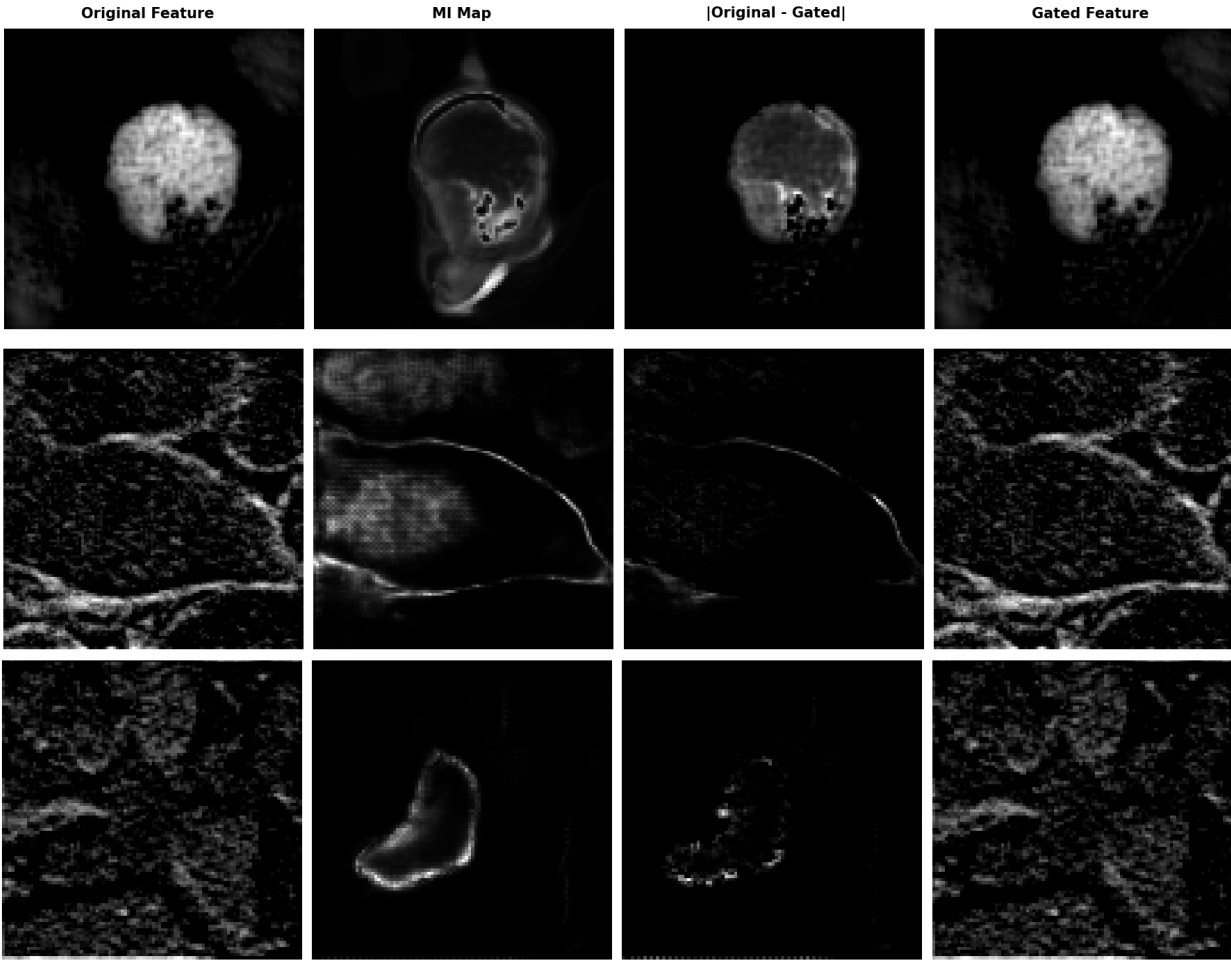

| Original Feature | MI Map | \|Original - Gated\| | Gated Feature |

Figure 1: Visualization of features before and after gating along with the absolute value of their difference. The first row is a sample from BraTS 2019, the second is from LA 2018, and the last is from NIH Pancreas-CT. It can be observed that the difference map follows the spatial structure of the MI map in regions where both uncertainty and feature activations are high, which is consistent with the implications of equation 7. Although the gated features remain visually similar to the original features due to the soft nature of the modulation, the difference maps reveal spatially selective attenuation patterns caused by the application of feedback gating.

$$|f_i - f_i \odot g_i| = |f_i - (f_i - f_i \odot u_i)| = |f_i \odot u_i| \tag{7}$$

Since the absolute difference is proportional to $|f_i \odot u_i|$, strong attenuation occurs primarily in regions where both the feature activations and uncertainty are high. Consequently, highly dominant but epistemically unreliable responses are selectively attenuated, reducing their influence on downstream segmentation predictions. In contrast, low-magnitude activations remain comparatively less affected even in uncertain regions, allowing the gating operation to preserve weaker contextual and semantic information rather than aggressively suppressing it through hard masking. This behavior is consistent with the visualizations in Figure 1.

It should also be noted that other methods of introducing uncertainty feedback, such as passing the MI maps as a second channel or introducing trainable weights, are also possible. However, we focus on keeping the parameters of the model constant (non-trainable additions) so as not to increase the computational requirements and memory of the model.

---

**Algorithm 1** Proposed training pipeline

---

1: **Input:** Given labeled data $D_{lab} = \{(x_i, y_i)|i \in [1, N_{lab}]\}$ and unlabeled data $D_{unlab} = \{x_i|i \in [1, N_{unlab}]\}$, compose $D$ consisting of batches $B$ containing batches of labeled and unlabeled data.

2: **Output:** $\theta$, the trained model weights

3: $\theta \leftarrow$ Initialize weights
4: **for** $iter = 1$ to $N_{max\_iterations}$ **do**
5:     Sample mini-batch $B \sim D$
6:     $(X_{lab}, Y_{lab}), X_{unlab} \leftarrow B$
7:     Generate weak and strong augmentations $(X_{weak}, X_{strong})$ from $X_{unlab}$
8:     Compute unsupervised weight $\lambda$ as per equation 13

9:     SUPERVISED STEP:
10:     Forward pass on $X_{lab}$ with deep supervision
11:     Compute supervised loss $\mathcal{L}_{sup}$ using equation 9

12:     UNSUPERVISED STEP:
13:     Perform $T$ stochastic forward passes on $X_{weak}$ with MC Dropout
14:     Compute pseudo-labels $\hat{Y}_{pseudo}$ as the mean of $N_{decoders} \times T$ outputs
15:     Compute the array of MI maps $MI = \{MI_1, \ldots, MI_{N_{decoders}}\}$ using equation 3
16:     $(\tilde{X}_{strong}, \tilde{\hat{Y}}_{pseudo}, \tilde{MI}) \leftarrow \text{CutMix3D}(X_{strong}, \hat{Y}_{pseudo}, MI)$
17:     Forward pass on $\tilde{X}_{strong}$ with MI-based feedback gating using $\tilde{MI}$
18:     Compute consistency loss $\mathcal{L}_{unsup}$ using equation 11

19:     $\mathcal{L} = \mathcal{L}_{sup} + \lambda \cdot \mathcal{L}_{unsup}$
20:     Update $\theta$ using backpropagation
21:     Update $lr$ using equation 14
22: **end for**
23: **return** $\theta$

---

## 3.4 Training Pipeline

We integrate the aforementioned components into a unified training pipeline. It consists of training iterations, and each iteration can be divided into 2 steps. These are the supervised and unsupervised training steps. Algorithm 1 gives the pseudo-code for the training pipeline.

### 3.4.1 Supervised Training

The supervised training step involves getting the prediction on the labeled input and computing the supervised loss component of the total loss. We compute the supervised loss as the 1:1 combination of dice $\mathcal{L}_{dice}$ loss and cross entropy $\mathcal{L}_{ce}$ loss between each decoder's output and the ground truth. The loss is then scaled by the total number of terms, which is two times the number of decoders. This is illustrated in equation 8 and equation 9, where $\hat{y}_i$ is the $i^{th}$ decoder's logits on the labeled input, $\hat{\mathcal{D}}_{lab}[\,i\,]$, passed through a softmax function over the class dimension.

$$\mathcal{L}_i = \mathcal{L}_{dice}(\hat{y}_i, Y_{lab}) + \mathcal{L}_{ce}(\hat{y}_i, Y_{lab}) \tag{8}$$

$$\mathcal{L}_{sup} = \frac{1}{2 \cdot N_{decoders}} \sum_{i=1}^{N_{decoders}} \mathcal{L}_i \tag{9}$$

### 3.4.2 Unsupervised Training

The unsupervised step involves the computation of the Mutual Information maps, followed by consistency regularization between the strong outputs generated by including the feedback gating mechanism and the pseudo-labels computed from the weak outputs.

We first perform a Monte Carlo Dropout operation with $T$ steps on the weak inputs, $X_{weak}$. Every $t^{th}$ step yields the final decoder's logits $\hat{y}_{weak}^t$ and the array of decoder logits $\hat{\mathcal{D}}_{weak}^t$. Using these $N_{decoders} \times T$ predictions, we compute the pseudo-labels $\hat{Y}_{pseudo}$ as described in equation 10.

$$\hat{Y}_{pseudo} = \frac{1}{N_{decoders} \cdot T} \sum_{t=1}^{T} \sum_{i=1}^{N_{decoders}} \text{softmax}\Big(\hat{\mathcal{D}}_{weak}^t[\,i\,]\Big) \tag{10}$$

For each decoder $i$, we compute the Mutual Information maps using the approximation in equation 3. Each $MI_i$ is clamped to range between 0 and 1.

We then apply 3D CutMix (Yun et al., 2019) to the strong inputs $X_{strong}$, pseudo-labels $\hat{Y}_{pseudo}$, and each of the MI maps to obtain their mixed versions $\tilde{X}_{strong}, \tilde{\hat{Y}}_{pseudo}$ and $\tilde{MI} = \{\tilde{MI}_1 \ldots \tilde{MI}_{N_{decoders}}\}$. The mixed MI maps, $\tilde{MI}$, are used to apply feedback gating during the forward pass on $\tilde{X}_{strong}$. The final unsupervised loss $\mathcal{L}_{unsup}$ is the sum of the Kullback-Leibler Divergence (Kullback & Leibler, 1951) between the softmax of each of the decoder logits from the strong input and $\tilde{\hat{Y}}_{pseudo}$ as described in equation 11.

$$\mathcal{L}_{unsup} = \frac{1}{N_{decoders}} \sum_{i=1}^{N_{decoders}} \mathcal{L}_{KL}\Big(\text{softmax}(\hat{\mathcal{D}}_{strong}[\,i\,]), \tilde{\hat{Y}}_{pseudo}\Big) \tag{11}$$

The semi-supervised loss is computed as:

$$\mathcal{L} = \mathcal{L}_{sup} + \lambda \cdot \mathcal{L}_{unsup} \tag{12}$$

where $\lambda$ is a sigmoid ramp-up factor (Tarvainen & Valpola, 2017), computed as described in equation 13.

$$\lambda = w_{consistency} \cdot \exp\left(-0.5 \cdot \left(1 - \frac{iter_{scaled}}{l_{rampup}}\right)^2\right) \tag{13}$$

where $iter_{scaled}$ is the current iteration number divided by a set iteration scaler, and $l_{rampup}$ is the ramp-up length. We choose these two values so that the value of $\lambda$ reaches $w_{consistency}$ by the end of training. The ramp-up phase allows the model to slowly incorporate the influence of unlabeled data, allowing the model to use the knowledge gained from the labeled data on the unlabeled data.

The learning rate for the optimizer follows a standard polynomial decay schedule and is updated as a function of the current iteration number ($iter$) as described in equation 14. $lr_o$ is the initial learning rate.

$$lr = lr_o \cdot (1 - \frac{iter}{\mathcal{N}_{max\_iterations}})^{0.9} \tag{14}$$

## 4 Experimentation and Results

### 4.1 Datasets

#### 4.1.1 BraTS 2019 Pre-Operative MRI Glioma Segmentation

The BraTS 2019 (Menze et al., 2014; Bakas et al., 2017; 2019) dataset is a widely used and publicly available dataset for training and benchmarking results related to Brain Tumor Segmentation. It consists of 259 MRI scans showing High-Grade Gliomas and 76 scans showing Low-Grade Gliomas, all of which are available in the four modalities (T1, T2, T1ce, and FLAIR). The provided ground truth segmentations contain three tumor subregions (excluding the background): Peritumoral Edema (PTE), Necrotic/Non-Enhancing Tumor Core (NCR/NET), and Enhancing Tumor (ET). Each scan is a 3D volume of dimensions $240 \times 240 \times 155$.

The dataset was split into training (250), validation (25), and testing (60) sets[1], with the training set further divided into labeled and unlabeled sets.

### 4.1.2 BraTS 2024 Post-Treatment Glioma Segmentation

The BraTS 2024 Post-Treatment Glioma (Karargyris et al., 2023; de Verdier et al., 2024) (BraTS-GLI 2024) dataset contains post-treatment MRI volumes for segmenting High Grade and Low Grade Gliomas. We use the publicly available training set, which contains 1350 samples, each of four modalities (T1, T2, T1ce, and FLAIR). The provided ground truths contain four tumor subregions (excluding the background): Surrounding Non-enhancing FLAIR Hyperintensity (SNFH), Non-Enhancing Tumor Core (NETC), Enhancing Tissue (ET), and the Resection Cavity (RC). Each scan is a 3D volume of dimensions $182 \times 218 \times 182$. Due to post-treatment effects and altered tissue appearance, this dataset represents a challenging, uncommonly tested benchmark compared to pre-operative BraTS datasets.

The dataset was randomly split into training (945), validation (135), and testing (270) sets, with the training set further divided into labeled and unlabeled sets.

### 4.1.3 Left Atrium Segmentation 2018

The 2018 Left Atrium Segmentation Challenge dataset (Xiong et al., 2021) is a very popular benchmark dataset for semi-supervised 3D medical image segmentation methods. It consists of 100 Late-Gadolinium Enhanced (LGE) MRI volumes, which we split into training (80) and testing (20) sets following (Yu et al., 2019), with the training set further divided into labeled and unlabeled subsets. The provided ground truths contain a single class corresponding to the Left Atrium.

### 4.1.4 NIH Pancreas-CT

The NIH Pancreas-CT dataset (Roth et al., 2016) consists of 82 contrast-enhanced abdominal CT scans and serves as another standard benchmark for semi-supervised 3D medical image segmentation methods. Each volume is annotated with a single foreground class corresponding to the pancreas. Following existing semi-supervised learning approaches (Peiris et al., 2023; Zhao et al., 2024; Wang et al., 2025), 20 volumes are reserved for testing, and the remaining scans are used for training with labeled and unlabeled splits.

## 4.2 Data Preprocessing

For BraTS 2019 and BraTS-GLI 2024, the samples were cropped ($128 \times 128 \times 128$) to eliminate the majority of the background voxels to improve the focus of the model towards the brain tissue. The scans are then min-max normalized to keep intensities uniform and preserve their relevance, followed by a z-normalization to ensure statistical consistency between the scans (Ranjbarzadeh et al., 2021), before stacking the desired modalities (in this case FLAIR and T1ce). For training, we extract random ($96 \times 96 \times 96$) patches following standard benchmarks. For inferencing, we use a sliding window approach with 50% overlap and Gaussian smoothing, using the MONAI (Cardoso et al., 2022) framework (version 1.4.0). While we have chosen FLAIR and T1ce for training, our approach is not limited to the use of a fixed number of modalities.

For the LA 2018 dataset, we perform a z-normalization of the volumes and extract random ($112 \times 112 \times 80$) patches following standard benchmarks. For inferencing, we follow (Yu et al., 2019) using a sliding window approach with a stride of $18 \times 18 \times 4$.

The Pancreas-CT volumes are preprocessed according to prior work (Peiris et al., 2023; Zhao et al., 2024; Wang et al., 2025) to ensure a fair comparison. During training, random ($96 \times 96 \times 96$) patches are extracted, while inference is performed using a sliding window strategy with a stride of $16 \times 16 \times 16$.

---

[1] https://github.com/HiLab-git/SSL4MIS/tree/master/data/BraTS2019 (Accessed July 28, 2025)

### 4.3 Augmentations

In the training pipeline, strong and weak augmentations are applied to the inputs for computing the mutual information maps and for computing the consistency loss. For weak augmentations, random 90-degree flips and rotations about the central axes were done to the samples for BraTS 2019, BraTS-GLI, and LA 2018 datasets. For Pancreas-CT, this was not done following common practice (Peiris et al., 2023; Zhao et al., 2024; Wang et al., 2025). For strong augmentations, Gaussian noise was added to the weak augmentations. The standard deviation of this noise was sampled uniformly from $[0.25, 0.75]$ at each iteration. For Pancreas-CT and LA 2018, random brightness and contrast perturbations are additionally applied before noise injection. Additionally, during the unsupervised training step, we apply 3D CutMix (Yun et al., 2019) to the strongly augmented inputs, corresponding pseudo-labels, and MI maps, with a probability of 1.0 and a beta parameter of 4.

### 4.4 Experimental Setup

All experiments were conducted under low-label regimes commonly used in semi-supervised 3D medical image segmentation. For BraTS 2019 and BraTS-GLI 2024, 25 and 50 labeled volumes were used. For LA 2018, experiments were performed with 8 and 16 labeled volumes, while for the Pancreas-CT dataset, 6 and 12 labeled volumes were used. $T$ was set to 5 during training and $N_{decoders}$ was set to 4. The SGD optimizer was used, with momentum set to 0.9. The learning rate was gradually decreased from 0.1 ($lr_o$) as a function of the iteration, and the L2 regularization parameter was set to 0.0001. Both labeled and unlabeled batch sizes were set to 2. The model was trained for a maximum of 30,000 iterations, with $l_{rampup}$ set to 150 and the iteration scaler set to 200, using mixed-precision on an NVIDIA RTX 4060 GPU.

### 4.5 Quantitative Comparison with Other Approaches

We evaluate the proposed method using standard segmentation metrics appropriate to each dataset.

For BraTS 2019 and BraTS-GLI 2024, performance is measured using the Dice Similarity Coefficient (DSC), Jaccard Index, and the 95th-percentile Hausdorff Distance (HD95). Following the BraTS evaluation protocol, metrics are computed over three clinically relevant composite regions: Whole Tumor (WT), Tumor Core (TC), and Enhancing Tumor (ET), as defined in equation 15 and equation 16. All methods are trained and evaluated using the same 3D UNet backbone, identical preprocessing, and identical inference settings to ensure a fair comparison.

$$\text{WT} = \begin{cases} \text{PTE} \cup \text{NCR/NET} \cup \text{ET}, & \text{if BraTS 2019,} \\ \text{SNFH} \cup \text{NETC} \cup \text{ET}, & \text{if BraTS-GLI 2024.} \end{cases} \tag{15}$$

$$\text{TC} = \begin{cases} \text{NCR/NET} \cup \text{ET}, & \text{if BraTS 2019,} \\ \text{NETC} \cup \text{ET}, & \text{if BraTS-GLI 2024.} \end{cases} \tag{16}$$

For the LA 2018 and Pancreas-CT datasets, we report DSC, Jaccard Index, HD95, and Average Surface Distance (ASD). All methods are trained and evaluated using a VNet backbone under the same preprocessing and inference protocols. For these two datasets, we directly quote results from the literature unless explicitly mentioned in the tables.

Quantitative results for BraTS 2019 and BraTS-GLI 2024 are summarized in Tables 2 and 3, respectively, while Tables 4 and 7 report results on LA 2018 and Pancreas-CT. Dice score distributions under different label regimes are visualized using boxplots in Figures 2, 4, and 6 to assess robustness and performance variability across test samples. In all tables, the best-performing results are highlighted in bold.

Table 2: Comparison of results on BraTS 2019

| Method | WT | | | TC | | | ET | | | Average * | | |
|---|---|---|---|---|---|---|---|---|---|---|---|---|
| | DSC (%, ↑) | Jac. (%, ↑) | HD95 (↓) | DSC (%, ↑) | Jac. (%, ↑) | HD95 (↓) | DSC (%, ↑) | Jac. (%, ↑) | HD95 (↓) | DSC (%, ↑) | Jac. (%, ↑) | HD95 (↓) |
| 25 (10%) Labeled Samples | | | | | | | | | | | | |
| 3D UNet (Çiçek et al., 2016) | 81.56 | 70.47 | 18.30 | 65.10 | 52.32 | 24.46 | 65.19 | 54.53 | 13.40 | 58.85 | 46.48 | 17.61 |
| UA-MT (Yu et al., 2019) | 80.37 | 69.31 | 15.64 | 68.06 | 55.64 | 18.35 | 63.71 | 52.45 | 12.26 | 60.31 | 48.13 | 15.00 |
| CPS (Chen et al., 2021) | 82.24 | 71.38 | 17.14 | 71.46 | 59.51 | 19.48 | 66.71 | 55.50 | 15.26 | 63.07 | 50.57 | 16.23 |
| DTC (Luo et al., 2021) | 82.96 | 72.25 | 22.48 | 69.96 | 57.43 | 30.12 | 65.24 | 54.11 | 16.48 | 62.55 | 50.24 | 22.38 |
| URPC (Luo et al., 2022b) | 83.92 | 73.99 | 13.96 | 76.16 | 65.38 | 15.88 | 67.80 | 57.81 | 11.63 | 66.02 | 53.98 | 12.64 |
| MCF (Wang et al., 2023) | 83.34 | 72.81 | 16.95 | 71.01 | 58.68 | 20.25 | 65.84 | 54.24 | 13.14 | 62.56 | 49.91 | 15.53 |
| AD-MT (Zhao et al., 2024) | 85.16 | 75.53 | 11.39 | 76.75 | 65.32 | 19.85 | 65.71 | 55.09 | 36.64 | 65.84 | 53.92 | 11.69 |
| Ours | **86.27** | **77.23** | **10.41** | **79.28** | **68.58** | **12.69** | **70.58** | **60.49** | **6.83** | **68.18** | **56.46** | **9.97** |
| 50 (20%) Labeled Samples | | | | | | | | | | | | |
| 3D UNet (Çiçek et al., 2016) | 84.94 | 75.16 | 11.39 | 71.99 | 60.54 | 13.46 | 67.01 | 56.75 | 9.21 | 63.54 | 51.70 | 11.58 |
| UA-MT (Yu et al., 2019) | 84.03 | 73.89 | 12.55 | 73.89 | 62.60 | 11.98 | 67.66 | 57.51 | 7.98 | 65.05 | 52.93 | 10.10 |
| CPS (Chen et al., 2021) | 85.45 | 75.69 | 12.17 | 76.76 | 65.72 | 12.82 | 68.16 | 57.91 | 7.99 | 65.99 | 53.83 | 10.99 |
| DTC (Luo et al., 2021) | 85.09 | 75.47 | 9.86 | 76.87 | 66.00 | 9.28 | 67.25 | 56.87 | 5.91 | 66.26 | 54.10 | 9.49 |
| URPC (Luo et al., 2022b) | 86.95 | 78.23 | 8.37 | 81.17 | 71.19 | 11.61 | 69.91 | 60.42 | 8.73 | 69.58 | 58.19 | 8.88 |
| MCF (Wang et al., 2023) | 84.98 | 75.18 | 10.00 | 75.17 | 63.86 | 10.38 | 66.99 | 56.21 | 7.42 | 64.71 | 52.34 | 9.36 |
| AD-MT (Zhao et al., 2024) | 86.83 | 77.73 | 9.53 | 78.26 | 67.28 | 11.13 | 68.17 | 57.99 | 7.15 | 67.44 | 55.83 | 9.75 |
| Ours | **88.07** | **79.66** | **8.37** | **83.03** | **73.16** | **7.37** | **71.16** | **61.56** | **5.40** | **70.89** | **59.35** | **7.64** |

* PTE, NCR/NET, and ET classes were used to compute the average

### 4.5.1 Results on BraTS 2019

From Table 2, it can be observed that the proposed method consistently achieves the highest segmentation performance across all three tumor subregions, as measured by both DSC, Jaccard Index, and HD95. With only 25 (10%) labeled samples, it improves mean DSC by 1.11% (WT), 2.53% (TC), and 2.78% (ET) over the second-best methods. Our method maintains its advantage when increasing the labeled set to 50 (20%), with improvements of 1.12%, 1.86%, and 1.25% for WT, TC, and ET, respectively.

Two-sided paired t-tests on DSC and Jaccard Index confirm that our proposed method establishes a clear performance advantage that is statistically significant across the majority of benchmarks ($p < 0.05$), with particularly robust gains ($p < 10^{-5}$) in the anatomically complex TC and ET regions. AD-MT and URPC maintain competitive parity in the WT region at 25 and 50 labeled samples, respectively.

Beyond mean performance, the proposed method exhibits reduced variability among test samples, compared to competing approaches, as shown in Figure 2. Using 25 (10%) labeled samples, it achieves not only a higher median DSC, but also a tighter Inter-Quartile Region (IQR) on all three regions. With 50 (20%) labeled samples, the proposed method still maintains a higher median on all three regions, with URPC achieving comparable but slightly less tight IQRs on all WT and TC.

These findings indicate that the proposed method delivers superior and more stable segmentation performance on the BraTS 2019 dataset across the WT, TC, and ET regions. To complete the analysis of results on the BraTS 2019 dataset, a visual comparison of the performance of the proposed approach and other methods is provided in Figure 3. These samples have been taken from the test set used to evaluate and compare all of the mentioned approaches and the proposed approach.

### 4.5.2 Results on BraTS-GLI 2024

From the quantitative results presented in Table 3, the proposed method achieves substantially higher segmentation performance than all competing approaches on the BraTS-GLI 2024 dataset as measured by the

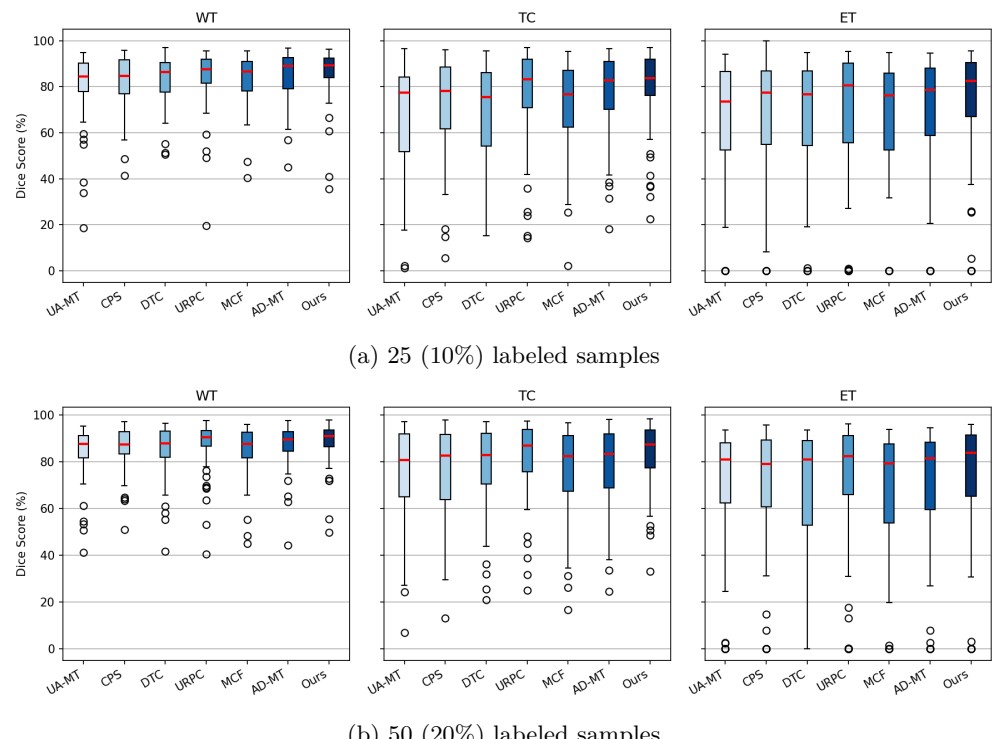

(a) 25 (10%) labeled samples

(b) 50 (20%) labeled samples

Figure 2: Dice score comparison on BraTS 2019

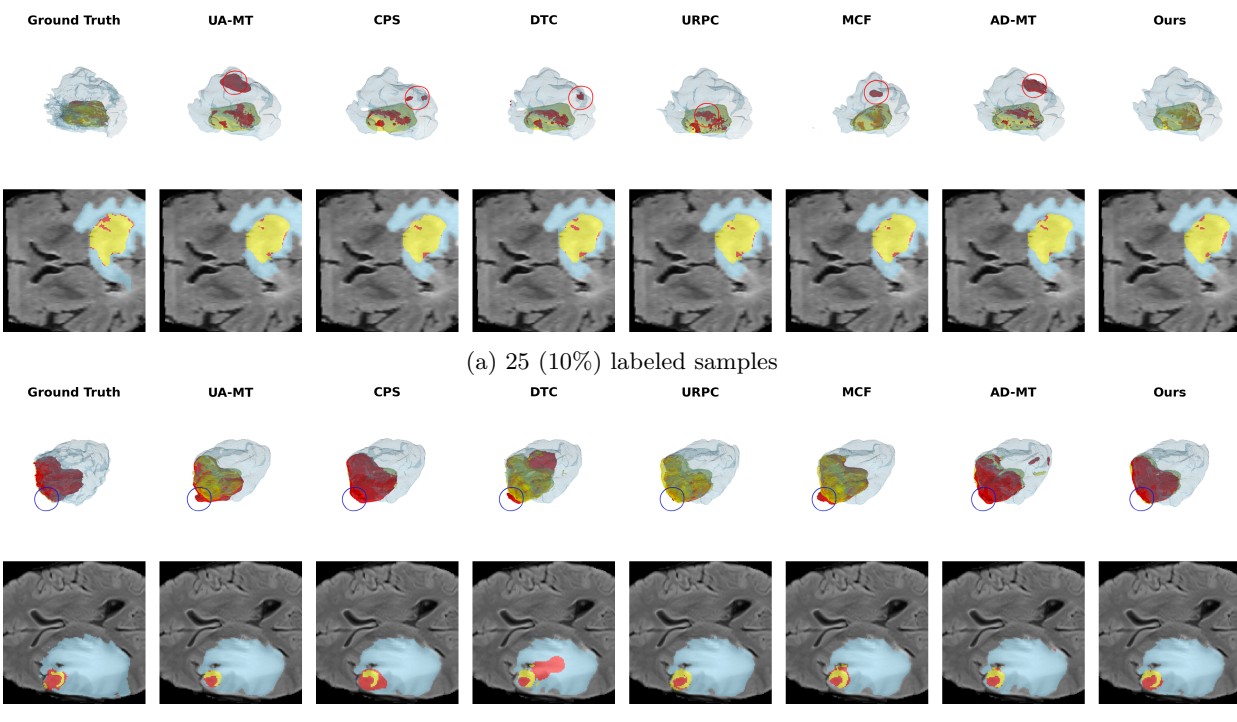

(a) 25 (10%) labeled samples

(b) 50 (20%) labeled samples

Figure 3: Visual comparison on BraTS 2019. The regions in blue, yellow, and red correspond to PTE, ET, and NCR/NET, respectively.

Table 3: Comparison of results on BraTS-GLI 2024

| Method | WT | | | TC | | | ET | | | Average * | | |
|---|---|---|---|---|---|---|---|---|---|---|---|---|
| | DSC (%, ↑) | Jac. (%, ↑) | HD95 (↓) | DSC (%, ↑) | Jac. (%, ↑) | HD95 (↓) | DSC (%, ↑) | Jac. (%, ↑) | HD95 (↓) | DSC (%, ↑) | Jac. (%, ↑) | HD95 (↓) |
| 25 (≈ 2.65%) Labeled Samples | | | | | | | | | | | | |
| 3D UNet (Çiçek et al., 2016) | 68.72 | 56.12 | 38.88 | 39.09 | 31.58 | 22.05 | 39.91 | 32.46 | 21.34 | 42.60 | 35.29 | 24.06 |
| UA-MT (Yu et al., 2019) | 70.12 | 56.68 | 27.76 | 39.53 | 32.38 | 15.45 | 39.05 | 31.94 | 14.36 | 48.61 | 40.87 | 17.74 |
| CPS (Chen et al., 2021) | 72.51 | 59.41 | 22.59 | 47.86 | 40.46 | 13.28 | 48.80 | 41.30 | 12.89 | 53.60 | 45.70 | 15.37 |
| DTC (Luo et al., 2021) | 73.83 | 61.35 | 21.39 | 40.46 | 33.11 | 15.55 | 42.37 | 34.97 | 15.28 | 47.33 | 39.28 | 16.61 |
| URPC (Luo et al., 2022b) | 79.57 | 68.61 | 18.81 | 53.63 | 46.77 | 9.49 | 54.30 | 47.37 | 8.54 | 57.31 | 49.84 | 13.40 |
| MCF (Wang et al., 2023) | 69.89 | 56.36 | 26.28 | 36.47 | 29.56 | 15.43 | 38.67 | 31.69 | 15.05 | 46.15 | 38.56 | 18.59 |
| AD-MT (Zhao et al., 2024) | 74.34 | 61.67 | 26.16 | 51.79 | 43.94 | 9.65 | 52.24 | 44.24 | 9.13 | 57.24 | 48.39 | 15.37 |
| Ours | **81.26** | **70.66** | **12.36** | **56.62** | **49.49** | **7.09** | **58.16** | **51.07** | **6.60** | **62.58** | **54.81** | **11.29** |
| 50 (≈ 5.30%) Labeled Samples | | | | | | | | | | | | |
| 3D UNet (Çiçek et al., 2016) | 72.63 | 60.69 | 33.74 | 40.24 | 33.44 | 26.35 | 40.26 | 33.55 | 25.44 | 46.04 | 38.41 | 26.12 |
| UA-MT (Yu et al., 2019) | 75.40 | 62.72 | 22.05 | 43.55 | 36.07 | 17.12 | 44.21 | 36.77 | 16.89 | 52.43 | 44.47 | 15.31 |
| CPS (Chen et al., 2021) | 78.51 | 66.67 | 17.90 | 45.77 | 38.07 | 14.02 | 46.16 | 38.37 | 13.04 | 56.11 | 47.57 | 13.88 |
| DTC (Luo et al., 2021) | 75.01 | 62.73 | 26.32 | 41.37 | 33.89 | 18.13 | 41.33 | 33.84 | 17.72 | 51.47 | 42.99 | 19.08 |
| URPC (Luo et al., 2022b) | 79.62 | 68.95 | **8.48** | 60.10 | 53.08 | **6.63** | 60.31 | 53.22 | **6.27** | 64.04 | 56.15 | **8.15** |
| MCF (Wang et al., 2023) | 73.49 | 61.13 | 19.26 | 48.73 | 41.08 | 13.46 | 48.28 | 40.47 | 12.94 | 52.29 | 43.70 | 15.00 |
| AD-MT (Zhao et al., 2024) | 76.06 | 63.69 | 21.99 | 43.54 | 35.99 | 13.95 | 44.72 | 37.18 | 13.59 | 49.12 | 40.79 | 17.47 |
| Ours | **82.42** | **72.40** | 11.72 | **61.12** | **53.99** | 7.08 | **61.22** | **54.03** | 6.72 | **64.45** | **56.51** | 9.63 |

\* SNFH, NETC, RC, and ET classes were used to compute the average

mean DSC, Jaccard Index, and HD95 on 25 labeled samples. The proposed approach shows improvements of 1.69% on WT, 2.99% on TC, and 3.86% on ET, as measured by the mean DSC, over the second-best methods. With 50 (≈ 5.30%) labeled samples, our approach shows improvements as measured by mean DSC and Jaccard Index, showing an increase of 2.80% on WT, 1.02% on TC, and 0.91% on ET. While the proposed approach falls behind in HD95 with 50 labeled samples, our performance on 25 samples indicates the effectiveness of the approach under limited labeled samples. The decreased gap in performance is expected as the number of labeled samples increases as approaches converge to a fully supervised setup.

Two-sided paired t-tests on DSC and Jaccard Index confirm that our proposed method establishes a clear performance advantage that is statistically significant across all benchmarks ($p < 0.05$) on WT, TC, and ET with 25 labeled samples. With 50 labeled samples, statistical significance is observed for all benchmarks on WT. On TC and ET, URPC maintains competitive parity.

Beyond improved mean performance, Figure 4 shows that the proposed approach not only has a higher median performance over the competing methods on all three regions, but also has a tighter Inter-Quartile Region (IQR), indicating more consistent predictions across the large test set of 270 volumes.

These results highlight the robustness of the proposed approach in the challenging post-treatment setting of BraTS-GLI 2024. Figure 5 presents a visual comparison between the proposed method and other approaches. The samples shown are drawn from the test set used to evaluate and compare all of the mentioned approaches and the proposed approach.

### 4.5.3 Results on LA 2018

From the quantitative results presented in Table 4, the proposed approach clearly outperforms other semi-supervised approaches when using 8 (10%) and 16 (20%) labeled samples as measured by mean DSC, Jaccard Index, HD95, and ASD. The proposed approach achieves improvements of 1.50% and 1.07% over the second-best methods as measured by DSC on 8 (10%) and 16 (20%) labeled, respectively.

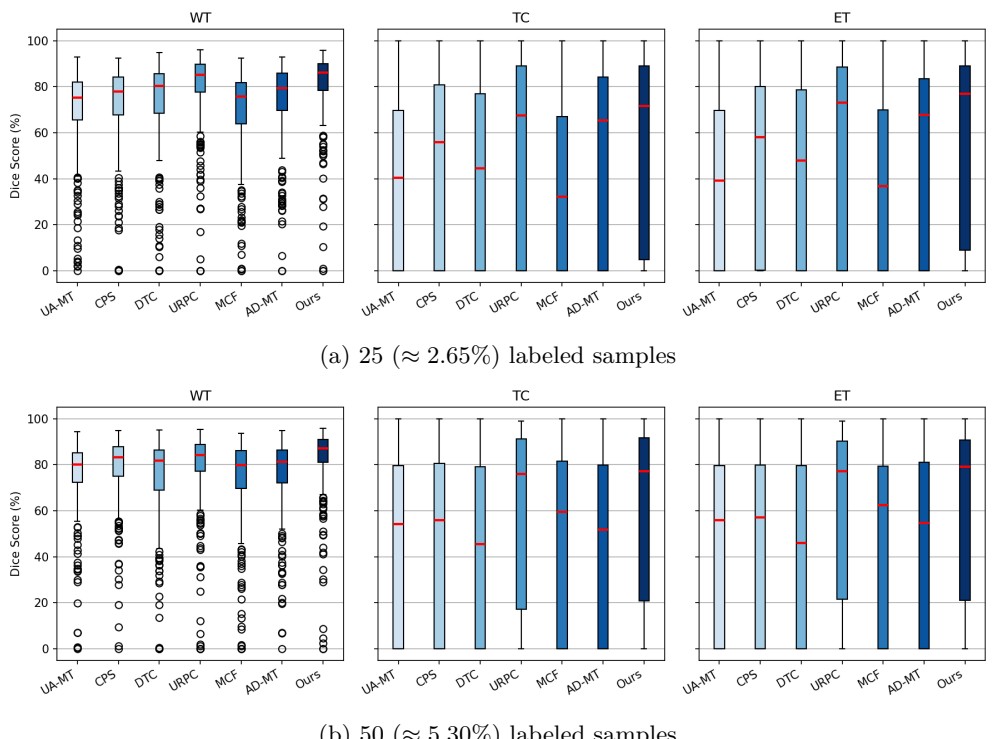

(a) 25 ($\approx 2.65\%$) labeled samples

(b) 50 ($\approx 5.30\%$) labeled samples

Figure 4: Dice score comparison on BraTS-GLI 2024

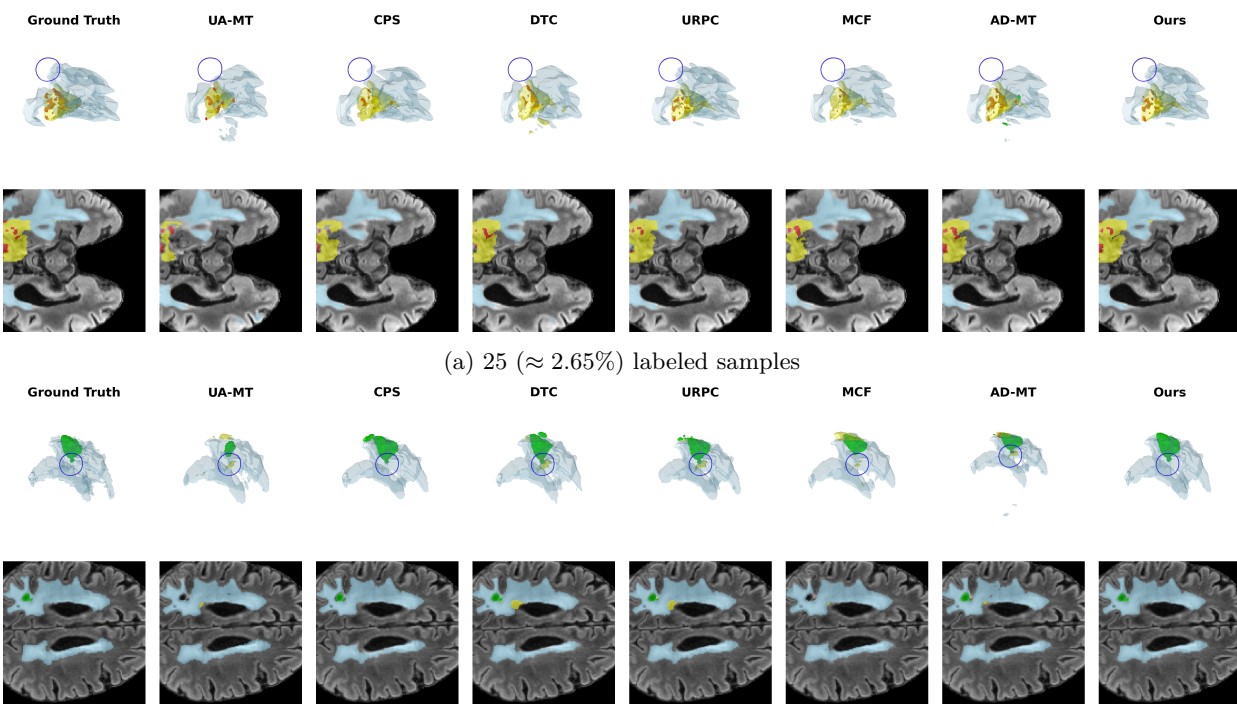

(a) 25 ($\approx 2.65\%$) labeled samples

(b) 50 ($\approx 5.30\%$) labeled samples

Figure 5: Visual comparison on BraTS-GLI 2024. The regions in blue, red, yellow, and green belong to the SNFH, NETC, ET, and RC classes, respectively.

Table 4: Comparison of results on LA 2018

| Method | 8 (10%) Labeled | | | | 16 (20%) Labeled | | | |
|---|---|---|---|---|---|---|---|---|
| | DSC ($\%, \uparrow$) | Jaccard ($\%, \uparrow$) | HD95 ($\downarrow$) | ASD ($\downarrow$) | DSC ($\%, \uparrow$) | Jaccard ($\%, \uparrow$) | HD95 ($\downarrow$) | ASD ($\downarrow$) |
| VNet* (Milletari et al., 2016) | 78.57 | 66.96 | 21.20 | 6.07 | 86.96 | 77.31 | 11.85 | 3.22 |
| UA-MT (Yu et al., 2019) | 83.37 | 71.99 | 17.91 | 4.41 | 87.68 | 78.36 | 14.39 | 3.52 |
| DTC (Luo et al., 2021) | 87.51 | 78.17 | 8.23 | 2.36 | 89.42 | 80.98 | 7.32 | 2.10 |
| URPC (Luo et al., 2022b) * | 85.02 | 75.98 | 15.21 | 2.95 | 88.43 | 81.15 | 8.21 | 2.35 |
| CAML (Gao et al., 2023) * | 89.62 | 81.28 | 8.76 | 2.02 | 90.78 | 83.19 | 6.11 | 1.68 |
| Co-BioNet (Peiris et al., 2023) [†] | 88.93 | 80.24 | 6.77 | 2.01 | 91.08 | 83.71 | 5.21 | 1.72 |
| AC-MT (Xu et al., 2023) [†] | 89.19 | 80.65 | 9.68 | 2.46 | 90.12 | 82.11 | 7.13 | 1.78 |
| AD-MT (Zhao et al., 2024) [†] | 90.05 | 81.97 | 6.67 | 1.67 | 91.40 | 84.23 | 5.96 | 1.49 |
| SGRS-Net (Wang et al., 2025) [†] | 90.27 | 82.35 | 6.82 | 1.67 | 90.80 | 83.27 | 6.12 | 1.94 |
| Ours | **91.77** | **84.84** | **5.13** | **1.44** | **92.47** | **86.05** | **4.43** | **1.33** |

\* Results quoted from CAML (Gao et al., 2023)
[†] Results obtained using the official implementation

Table 5: Comparison of results on LA 2018 across multiple seeds

| Method | DSC ($\%, \uparrow$) | Jaccard ($\%, \uparrow$) | HD95 ($\downarrow$) | ASD ($\downarrow$) |
|---|---|---|---|---|
| 8 (10%) Labeled | | | | |
| AC-MT (Xu et al., 2023) [†] | $88.04 \pm 0.81$ | $78.9 \pm 1.24$ | $9.22 \pm 0.38$ | $2.15 \pm 0.23$ |
| AD-MT (Zhao et al., 2024) [†] | $90.30 \pm 0.30$ | $82.46 \pm 0.55$ | $6.41 \pm 0.43$ | $1.61 \pm 0.04$ |
| SGRS-Net (Wang et al., 2025) [†] | $90.48 \pm 0.16$ | $82.7 \pm 0.26$ | $6.52 \pm 0.26$ | $1.70 \pm 0.07$ |
| Ours | $\mathbf{91.63 \pm 0.10}$ | $\mathbf{84.61 \pm 0.17}$ | $\mathbf{5.26 \pm 0.12}$ | $\mathbf{1.45 \pm 0.02}$ |
| 16 (20%) Labeled | | | | |
| AC-MT (Xu et al., 2023) [†] | $90.07 \pm 0.26$ | $82.05 \pm 0.42$ | $6.89 \pm 0.29$ | $1.69 \pm 0.08$ |
| AD-MT (Zhao et al., 2024) [†] | $91.44 \pm 0.11$ | $84.28 \pm 0.19$ | $5.72 \pm 0.27$ | $1.51 \pm \mathbf{0.02}$ |
| SGRS-Net (Wang et al., 2025) [†] | $90.42 \pm 0.42$ | $82.64 \pm 0.68$ | $6.38 \pm 0.62$ | $1.94 \pm 0.06$ |
| Ours | $\mathbf{92.41 \pm 0.09}$ | $\mathbf{85.94 \pm 0.15}$ | $\mathbf{4.41 \pm 0.17}$ | $\mathbf{1.37} \pm 0.07$ |

[†] Results obtained using the official implementation.

We also compare segmentation performance across three independent seeds $\{0, 42, 1337\}$, on AC-MT, AD-MT, SGRS-Net and our approach, using 8 (10%) and 16 (20%) labeled samples. The mean and standard deviation across the seeds for DSC, Jaccard Index, HD95 and ASD are reported in Table 5. The results from this table show that the proposed method not only achieves the best mean performance, but also exhibits consistently low variance across seeds.

To provide a more rigorous analysis, Figure 6 presents boxplots of Dice scores across all test cases. Under both 8 and 16 labeled settings, our method achieves the highest median Dice score. It can also be seen that the proposed approach has the tightest IQR, indicating low variability among test samples. Furthermore, two-sided paired t-tests against Co-BioNet, AC-MT, AD-MT, and SGRS-Net on DSC and Jaccard Index show that the improvements are statistically significant ($p < 0.05$) for all baselines at both 8 and 16 labeled samples, supporting the robustness of the observed gains. These findings, along with the analysis across different seeds together show that our approach show consistently low variability across test samples, and across test runs, which are complementary notions of robustness.

To assess computational efficiency, we compare GPU memory consumption and normalized training times against recent semi-supervised methods on the LA 2018 dataset using 8 (10%) labeled samples (Table 6). GPU memory usage was measured during training using `nvidia-smi`. Since different implementations define

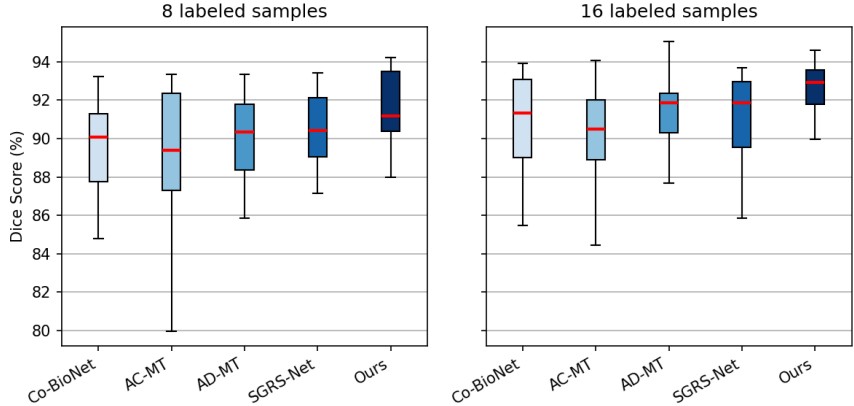

Figure 6: Dice score comparison on LA 2018

Table 6: Comparison of GPU memory consumption and normalized training times on LA 2018 with 8 (10%) labeled samples

| Method | GPU Memory (MiB) | Training Time (s/4 iterations) |
|---|---|---|
| Co-BioNet (Peiris et al., 2023) | 25266 [†] | − |
| AC-MT (Xu et al., 2023) | 5691 | $6.53 \pm 0.26$ |
| AD-MT (Zhao et al., 2024) | 6935 | $4.97 \pm 0.32$ |
| SGRS-Net (Wang et al., 2025) | 7445 | $4.14 \pm 0.08$ |
| Ours (without MP*) | 5445 | $4.85 \pm 0.08$ |
| Ours (with MP*) | **3401** | **2.32 ± 0.04** |

[†] Peak GPU memory consumption

[*] Mixed-Precision

epochs differently, we normalized the measurements to a fixed unit corresponding to 4 training iterations for fair comparison. All methods were evaluated on an NVIDIA RTX 4060 8GB GPU, except Co-BioNet, whose reported value was obtained on an NVIDIA Tesla V100 32GB GPU. The proposed method exhibits the lowest memory footprint among all compared approaches and remains memory efficient even without mixed-precision training. It also shows competitive training times without mixed-precision, while using additional stochastic forward passes.

To complete the analysis of results on the LA 2018 dataset, Figure 7 provides qualitative visual comparisons between the proposed method and other approaches on samples taken from the test set. The visual results highlight the proposed method's ability to capture fine structural details while maintaining smooth and consistent boundaries, which is consistent with the observed improvements in the ASD metric.

### 4.5.4 Results on Pancreas-CT

From the quantitative results presented in Table 7, the proposed method achieves competitive performance across all evaluation metrics on the Pancreas-CT dataset, outperforming recent state-of-the-art methods on both 6 (10%) and 12 (20%) labeled samples on mean DSC, Jaccard Index, and ASD. These results indicate that the proposed framework generalizes well beyond brain MRI to 3D CT organ segmentation tasks, despite the substantial differences in imaging modality and anatomical structure.

To conclude the analysis of results on the Pancreas-CT dataset, Figure 8 provides a visual comparison between the proposed approach and other approaches on samples taken from the test set. These qualitative results further validate the competitive performance achieved by our approach.

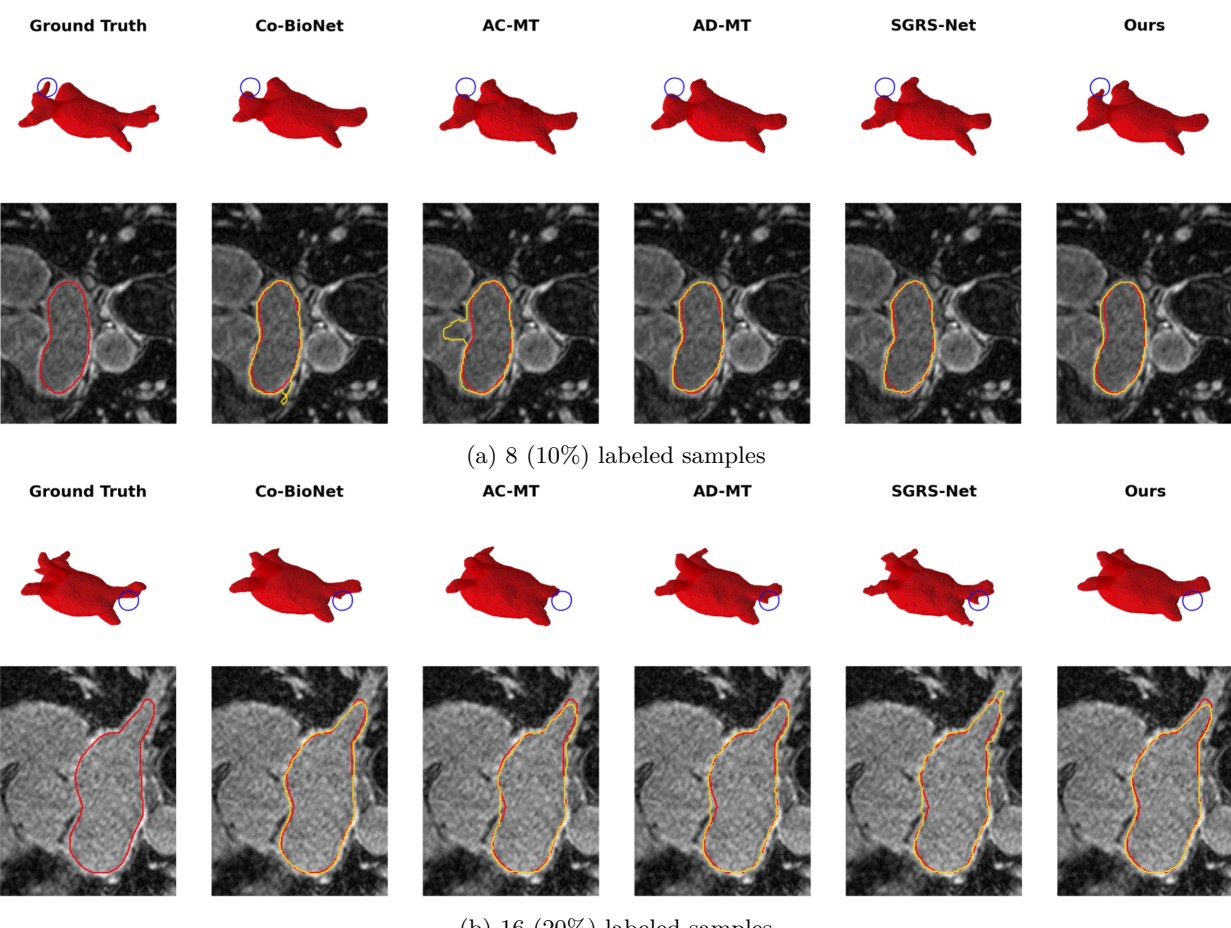

(a) 8 (10%) labeled samples

(b) 16 (20%) labeled samples

Figure 7: Visual comparison on LA 2018. The red contours indicate the Ground Truth, while the yellow contours represent the prediction.

Table 7: Comparison of results on Pancreas-CT

| Method | 8 (10%) Labeled | | | | 16 (20%) Labeled | | | |
|---|---|---|---|---|---|---|---|---|
| | DSC (%, ↑) | Jaccard (%, ↑) | HD95 (↓) | ASD (↓) | DSC (%, ↑) | Jaccard (%, ↑) | HD95 (↓) | ASD (↓) |
| VNet* (Milletari et al., 2016) | 55.06 | 40.48 | 32.80 | 12.67 | 69.65 | 55.19 | 20.20 | 6.31 |
| UA-MT (Yu et al., 2019) * | 68.71 | 54.65 | 13.89 | 3.23 | 78.17 | 65.22 | 6.9 | 1.55 |
| DTC (Luo et al., 2021) * | 66.27 | 52.07 | 15.00 | 4.44 | 76.75 | 63.77 | 8.52 | 1.72 |
| URPC (Luo et al., 2022b) | 74.89 | 59.85 | 11.30 | 3.74 | 80.31 | 67.09 | 6.58 | 2.10 |
| MCF (Wang et al., 2023) | – | – | – | – | 75.00 | 61.27 | 11.59 | 3.27 |
| Co-BioNet (Peiris et al., 2023) * | 77.89 | 64.79 | 8.81 | 1.39 | 82.22 | 70.24 | 7.71 | 1.16 |
| AD-MT (Zhao et al., 2024) | 80.21 | 67.51 | 7.18 | 1.66 | 82.61 | 70.70 | **4.94** | 1.38 |
| SGRS-Net (Wang et al., 2025) † | 78.74 | 65.48 | 14.07 | 5.18 | 81.30 | 68.86 | 5.88 | 1.48 |
| Ours | **80.93** | **68.42** | **6.81** | **1.17** | **82.77** | **70.94** | 6.19 | **1.10** |

* Results quoted from Co-BioNet (Peiris et al., 2023)
† Results obtained using the official implementation

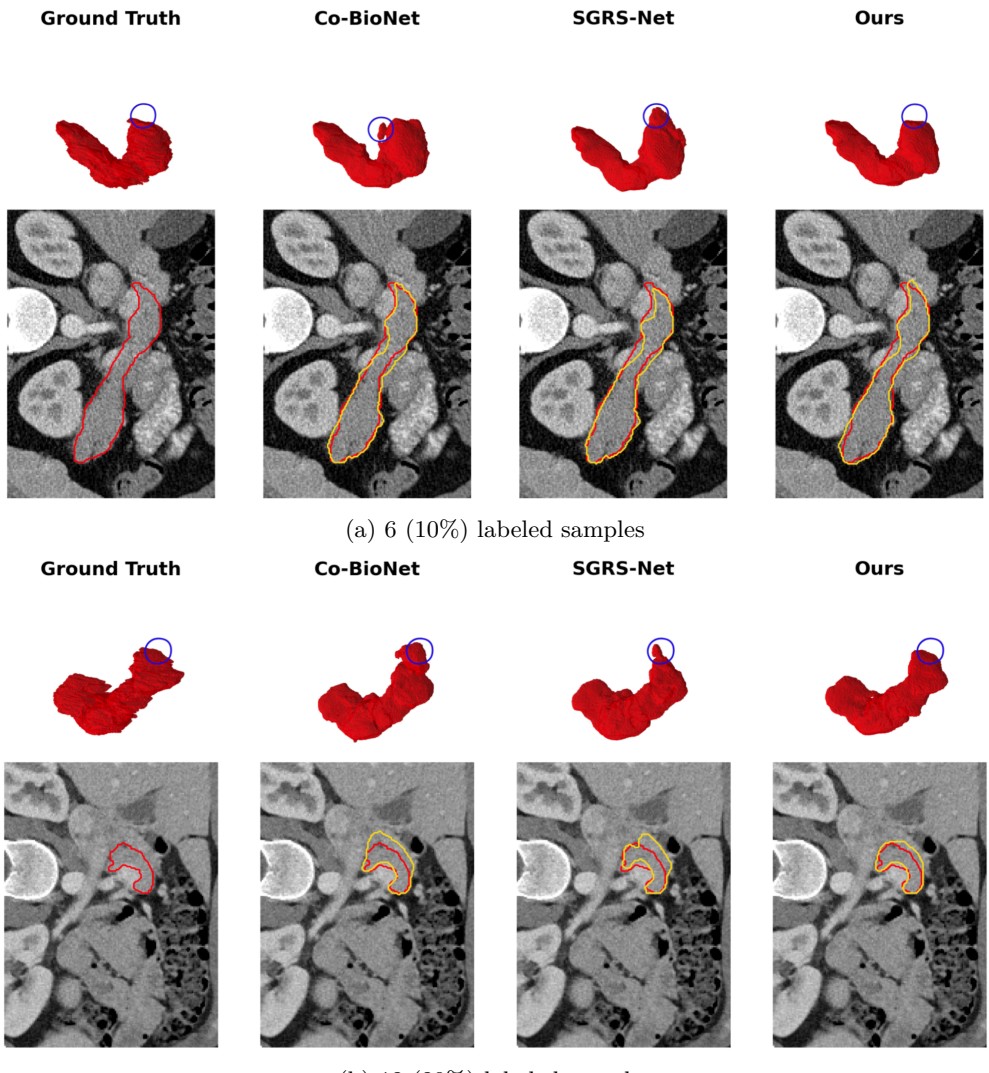

(a) 6 (10%) labeled samples

(b) 12 (20%) labeled samples

Figure 8: Visual comparison on Pancreas-CT. The red contours indicate the Ground Truth, while the yellow contours represent the prediction.

## 4.6 Ablation Study

### 4.6.1 On the Effect of Deep Supervision

To study the effect of deep supervision and, by extension, the omission of MI feedback to different decoder blocks, we include each scale of deep supervision incrementally with the total number of decoder blocks, $N_{decoders}$ set to 4. The omission of deep supervision also implies the removal of MI feedback for that particular decoder block, since the MI maps for a block are computed from the MC Dropout outputs from the deep supervision head of that block. We conduct this study on the LA 2018 and Pancreas-CT datasets using the VNet backbone, and the results are shown in Table 8. It should be noted that $\hat{\mathcal{D}}_1$ denotes the final or highest resolution.

The results show that the complete use of deep supervision is beneficial in our context. It consistently gives the highest mean segmentation performance across all four metrics on LA 2018 and Pancreas-CT datasets. On Pancreas-CT, it is observed that the difference between using 2 and 4 levels of deep supervision doesn't

Table 8: Effect of Deep Supervision on LA 2018 and Pancreas-CT with 10% labeled samples

| $\hat{\mathcal{D}}_1$ | $\hat{\mathcal{D}}_2$ | $\hat{\mathcal{D}}_3$ | $\hat{\mathcal{D}}_4$ | DSC (%, ↑) | Jaccard (%, ↑) | HD95 (↓) | ASD (↓) |
|---|---|---|---|---|---|---|---|
| | | | | LA 2018 with 8 (10%) Labeled Samples | | | |
| ✓ | | | | 91.02 | 83.59 | 6.36 | 1.48 |
| ✓ | ✓ | | | 90.65 | 82.97 | 7.14 | 1.46 |
| ✓ | ✓ | ✓ | | 91.11 | 83.75 | 5.63 | 1.50 |
| ✓ | ✓ | ✓ | ✓ | **91.77** | **84.84** | **5.13** | **1.44** |
| | | | | Pancreas-CT with 6 (10%) Labeled Samples | | | |
| ✓ | | | | 79.41 | 66.53 | 8.23 | 1.25 |
| ✓ | ✓ | | | 80.87 | 68.34 | 6.99 | 1.21 |
| ✓ | ✓ | ✓ | | 79.98 | 67.18 | 7.62 | 1.29 |
| ✓ | ✓ | ✓ | ✓ | **80.93** | **68.42** | **6.81** | **1.17** |

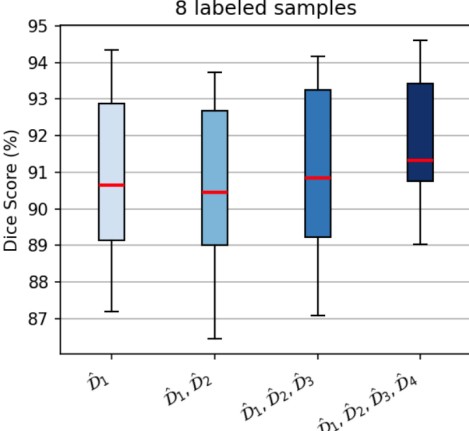

Figure 9: Dice score comparison between different levels of deep supervision on LA 2018

provide a statistically significant ($p < 0.05$) increase, whereas for LA 2018, using 4 levels of deep supervision provides statistically significant increases ($p < 0.05$) over the remaining three settings.

Figure 9 provides the boxplots of the Dice scores across all test cases on the LA 2018 dataset. It is clear from the figure that using all 4 levels of deep supervision not only results in a higher median, but also a tighter IQR.

### 4.6.2 On the Effect of 3D CutMix for Strong-Weak Consistency

To study the effect of 3D CutMix in our approach, we vary the CutMix probability, $P_{CutMix3D}$, from 0.0 to 1.0. We use the LA 2018 and Pancreas-CT datasets for this study with the VNet backbone. The results of this study are presented in Table 9.

The results show that increasing the use of 3D CutMix improves mean segmentation performance in our context. The most significant increase is shown in Pancreas-CT, with a 2.12% increase in mean DSC from complete omission, $P_{CutMix3D} = 0.0$ to $P_{CutMix3D} = 1.0$. Improvements are seen across all four metrics, with the highest being achieved when $P_{CutMix3D} = 1.0$.

Table 9: Ablation of 3D CutMix on LA 2018 and Pancreas-CT with 10% labeled samples

| $P_{CutMix3D}$ | DSC (%, ↑) | Jaccard (%, ↑) | HD95 (↓) | ASD (↓) |
|---|---|---|---|---|
| LA 2018 with 8 (10%) Labeled Samples | | | | |
| 0.0 | 91.01 | 83.58 | 6.11 | 1.54 |
| 0.5 | 91.50 | 84.39 | 5.45 | 1.47 |
| 1.0 | **91.77** | **84.84** | **5.13** | **1.44** |
| Pancreas-CT with 6 (10%) Labeled Samples | | | | |
| 0.0 | 78.81 | 65.67 | 8.54 | 1.34 |
| 0.5 | 78.99 | 66.08 | 9.12 | 1.31 |
| 1.0 | **80.93** | **68.42** | **6.81** | **1.17** |

Table 10: Results for varying $T$ on LA 2018 with 8 (10%) labeled samples

| $T$ | DSC (%, ↑) | Jaccard (%, ↑) | HD95 (↓) | ASD (↓) | Training Time (s/ep) * | GPU Memory (MiB) |
|---|---|---|---|---|---|---|
| 2 | 91.60 | 84.56 | 5.41 | 1.47 | **2.02 ± 0.07** | **3049** |
| 5 | **91.77** | **84.84** | **5.13** | **1.44** | 2.32 ± 0.04 | 3401 |
| 8 | 91.25 | 84.00 | 5.60 | 1.51 | 2.68 ± 0.06 | 3747 |

\* The training time is measured in seconds per epoch (s/ep). In this context, 1 epoch is equal to 4 iterations since we use a labeled batch size of 2, with 8 labeled samples

### 4.6.3 On the Impact of Stochastic Passes ($T$) on Performance and Efficiency

To study the effect of varying $T$ on segmentation performance and training efficiency, we vary the value of $T \in \{2, 5, 8\}$, keeping all other parameters constant, on the LA 2018 dataset using the VNet backbone. The results of this study are consolidated in Table 10. All experiments conducted as a part of this study were done on an NVIDIA RTX 4060 8GB GPU, and training times were recorded after sufficient warm-up and stabilization.

A performance peak is observed at $T = 5$ as measured by mean DSC, Jaccard Index, HD95, and ASD. Interestingly, increasing $T$ to 8 slightly degrades segmentation metrics, suggesting that an overly restrictive feedback gate may suppress useful learning signals. Furthermore, the computational overhead of increasing $T$ is minimal. $T = 5$ requires only 0.3s more per epoch than $T = 2$, with a 352 MiB increase in memory, which is significantly lower than the introduction of an additional model instance. These results justify our selection of $T = 5$ as an optimal balance between uncertainty precision and training efficiency.

### 4.6.4 On the Placement of Mutual Information Feedback Gating

To study the effect of where MI feedback should be injected, we evaluate two plausible gating locations within the decoder: encoder features (skip connections) and decoder features, as illustrated in Figure 10. These locations, labeled 1 and 2 in Figure 10, correspond to gating encoder features before fusion with decoder activations and gating decoder representations directly before deep-supervision, respectively. We conduct this analysis on BraTS 2019 and Pancreas-CT using 10% labeled data. All studies on the BraTS 2019 were conducted using the 3D UNet architecture, while the Pancreas-CT studies used the VNet.

The results in Tables 11 and 12 reveal that the optimal placement of MI feedback is architecture- and dataset-dependent. For the 3D UNet backbone used in BraTS 2019, applying MI feedback to decoder features consistently yields superior performance across all tumor subregions, while for the VNet backbone used in Pancreas-CT, encoder gating clearly outperforms decoder gating. We hypothesize this may be related to how skip-connections are integrated (concatenation vs. residual connection), though a formal investigation of this architectural interaction is left for future work. Accordingly, we adopt location 2 for the 3D UNet and location 1 for the VNet as the final configuration.

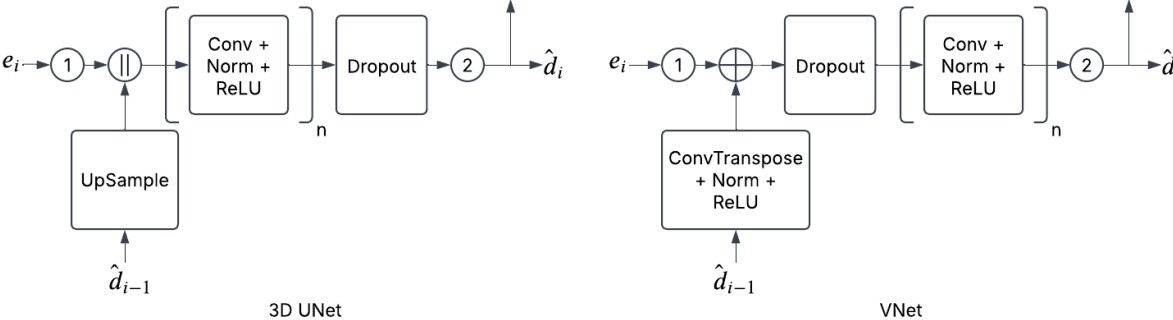

Figure 10: Schematic diagram of a single decoder block showing the two gating locations (labeled 1 and 2) for both the 3D UNet and VNet architectures

Table 11: Comparison between MI Feedback applied on two gating locations on BraTS 2019 with 25 (10%) labeled samples

| Region | Location | DSC (%) (↑) | Jaccard (%) (↑) | HD95 (↓) |
|---|---|---|---|---|
| WT | 1 | 84.13 | 74.64 | 11.23 |
|  | 2 | **86.27** | **77.23** | **10.41** |
| TC | 1 | 77.15 | 66.32 | 13.05 |
|  | 2 | **79.28** | **68.58** | **12.69** |
| ET | 1 | 70.03 | 60.16 | 7.24 |
|  | 2 | **70.58** | **60.49** | **6.83** |
| Average * | 1 | 67.50 | 55.74 | 10.33 |
|  | 2 | **68.18** | **56.46** | **9.97** |

\* PTE, NCR/NET, and ET classes were used to compute the average

### 4.6.5 On the Effect of Mutual Information Feedback Gating

To experimentally validate the effect of MI feedback during training, we compare the performance of the approach with and without MI feedback while keeping all other parameters constant on BraTS 2019 and Pancreas-CT datasets, using 10% labeled samples. These results are consolidated in Tables 13 and 14.

The results in Table 13 show an overall benefit to using MI feedback on the BraTS 2019 dataset with 10% labeled samples, when using the 3D UNet. The benefits are marginal on the WT region but are more profound on the TC and ET classes, with a considerable 2.19% increase in mean DSC on ET. A more elaborate analysis in Figure 11 reveals visibly tighter IQR on the TC and ET regions, indicating improved consistency on test samples when using MI feedback. These results suggest that MI feedback is especially beneficial for regions with complex boundaries or small spatial extent, such as the TC and ET regions in BraTS 2019. In contrast, larger and more prominent structures, such as WT, exhibit smaller gains.

Table 12: Comparison between MI Feedback applied on two gating locations on Pancreas-CT with 6 (10%) labeled samples

| Location | DSC (%, ↑) | Jaccard (%, ↑) | HD95 (↓) | ASD (↓) |
|---|---|---|---|---|
| 1 | **80.93** | **68.42** | **6.81** | **1.17** |
| 2 | 79.67 | 66.77 | 7.99 | 1.30 |

Table 13: Ablation of MI Feedback on BraTS 2019 with 25 (10%) labeled samples

| Region | MI | DSC (%) (↑) | Jaccard (%) (↑) | HD95 (↓) |
|--------|-----|-------------|-----------------|----------|
| WT | × | 86.02 | 77.15 | **10.11** |
| | ✓ | **86.27** | **77.23** | 10.41 |
| TC | × | 78.41 | 67.77 | 15.27 |
| | ✓ | **79.28** | **68.58** | **12.69** |
| ET | × | 68.39 | 58.58 | 9.75 |
| | ✓ | **70.58** | **60.49** | **6.83** |
| Average * | × | 67.70 | 55.96 | 10.78 |
| | ✓ | **68.18** | **56.46** | **9.97** |

* PTE, NCR/NET, and ET classes were used to compute the average

Table 14: Ablation of MI Feedback on Pancreas-CT with 6 (10%) Labeled Samples

| MI | DSC (%, ↑) | Jaccard (%, ↑) | HD95 (↓) | ASD (↓) |
|-----|------------|-----------------|----------|---------|
| × | 80.31 | 67.54 | 7.40 | 1.38 |
| ✓ | **80.93** | **68.42** | **6.81** | **1.17** |

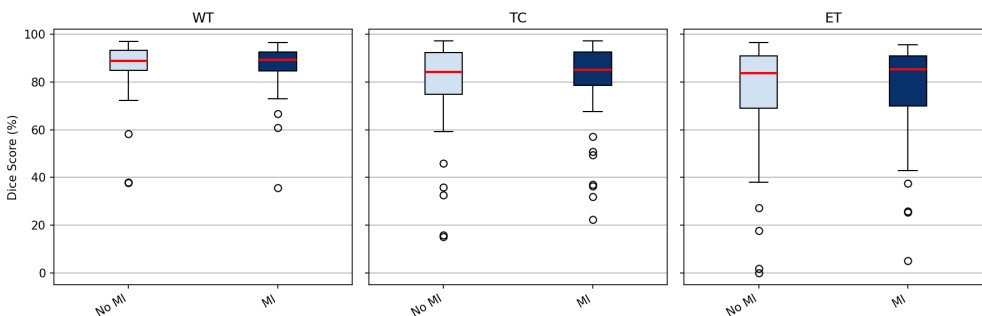

Figure 11: Dice score comparison between MI and no MI on BraTS 2019 with 25 (10%) labeled samples

From Table 14, we observe better performance in the presence of MI feedback on Pancreas-CT as measured by all four metrics.

To study the impact of MI feedback on clinical utility, we evaluate model robustness and probabilistic reliability. We subject the BraTS 2019 (MRI) models to varying levels of Rician noise and the Pancreas-CT models to Quantum Mottle, measuring the stability of DSC. We model Quantum Mottle as a scaled Poisson process, as illustrated in equation 17. In this equation, $\mathcal{P}(\lambda)$ represents a Poisson distribution with $\lambda$ as the mean, $N$ represents the simulated photon flux, $x$ is the input volume and $y$ is the noised volume. Under this model, the variance is $\frac{x-x_{min}}{N}$.

$$y = \frac{1}{N} \cdot \mathcal{P}(N \cdot (x - x_{min})) + x_{min} \tag{17}$$

Additionally, we generate reliability diagrams to visualize the alignment between voxel-wise accuracy and confidence probability. These results are detailed in Figure 12 and 13.

From both figures, it can be observed that the presence of MI feedback during training yields substantial improvement in model calibration. We observe a consistent decrease in Expected Calibration Error (ECE) across all experiments. Furthermore, we can observe that the presence of MI improves robustness to noise,

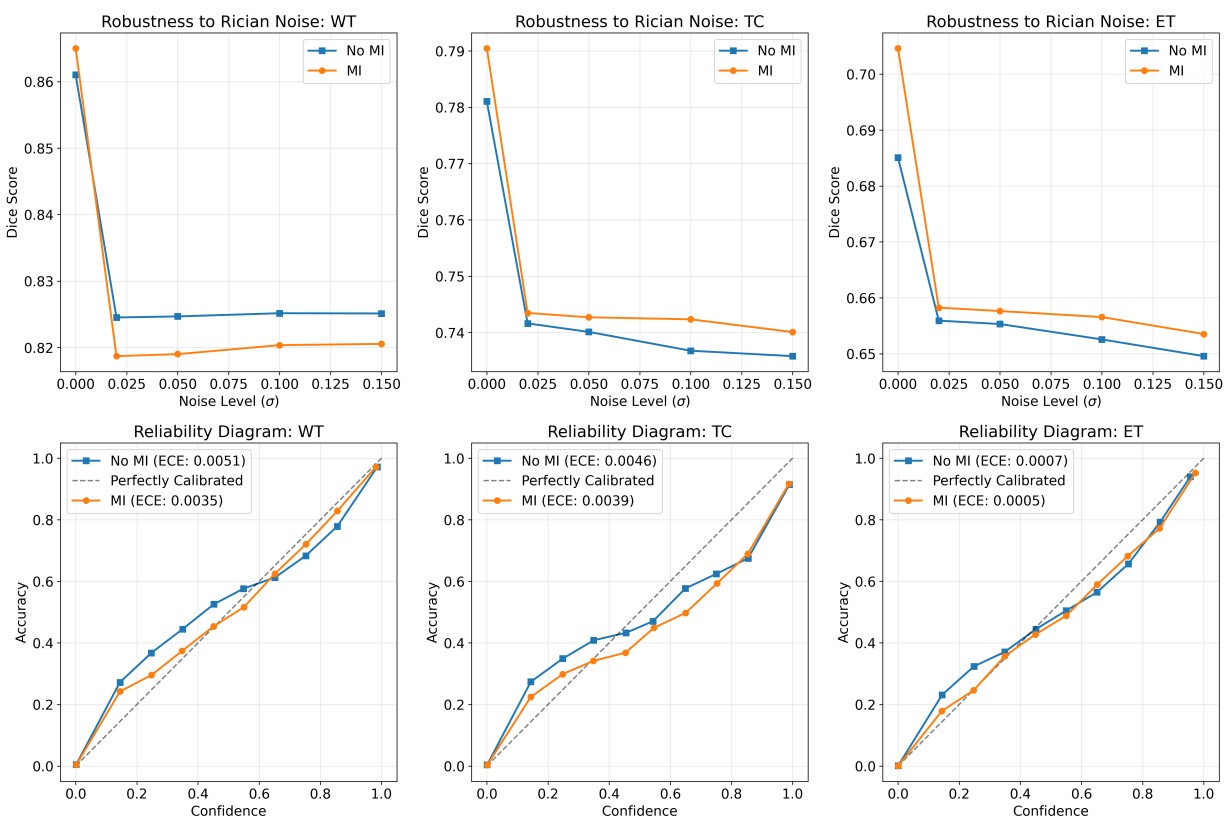

Figure 12: Robustness to Rician Noise and Reliability Diagram with and without MI on BraTS 2019 using 25 (10%) labeled samples

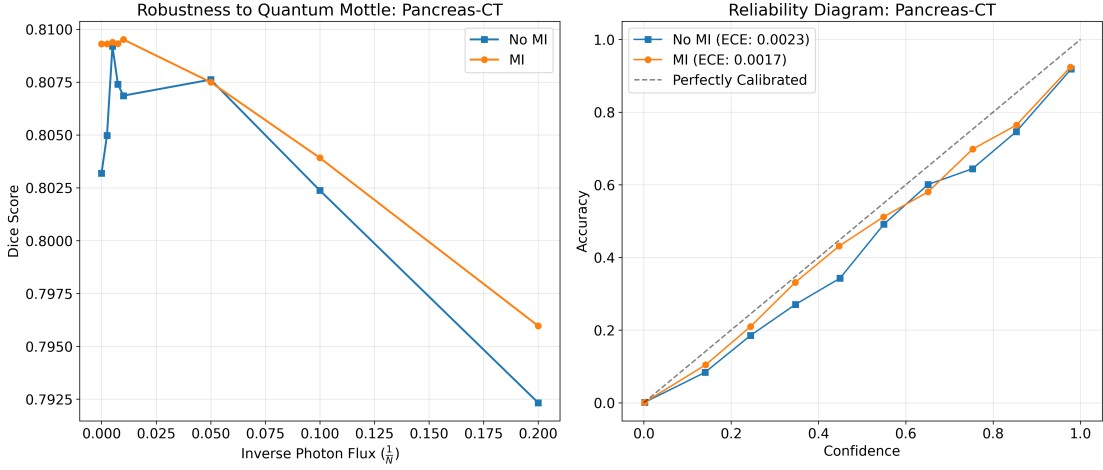

Figure 13: Robustness to Quantum Mottle and Reliability Diagram with and without MI on Pancreas-CT using 6 (10%) labeled samples

particularly in regions that are inherently complex to segment, such as the TC and ET regions in BraTS 2019, as well as the Pancreas in Pancreas-CT. Interestingly, in the case of Pancreas-CT, we observe an increase in performance at moderate flux levels ($1/N \leq 0.01$) before a steady decline at the start of photon starvation, with MI showing better stability. While the absence of MI feedback during training maintains a

Table 15: Predictive Entropy ($\mathcal{H}$) vs. MI on BraTS 2019 with 25 (10%) labeled samples

| Region | MI / $\mathcal{H}$ | DSC (%, ↑) | Jaccard (%, ↑) | HD95 (↓) |
|---|---|---|---|---|
| WT | $\mathcal{H}$ | 85.65 | 76.41 | 11.70 |
| | MI | **86.27** | **77.23** | **10.41** |
| TC | $\mathcal{H}$ | 78.11 | 67.35 | 13.17 |
| | MI | **79.28** | **68.58** | **12.69** |
| ET | $\mathcal{H}$ | 68.97 | 58.91 | 8.91 |
| | MI | **70.58** | **60.49** | **6.83** |
| Average * | $\mathcal{H}$ | 67.64 | 55.82 | 10.68 |
| | MI | **68.18** | **56.46** | **9.97** |

* PTE, NCR/NET, and ET classes were used to compute the average

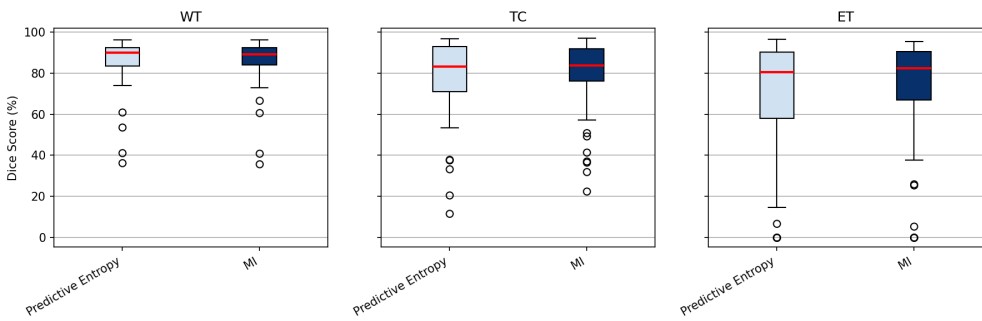

Figure 14: Dice score comparison between MI and Predictive Entropy

marginal DSC advantage in the high-volume WT region, this is overshadowed by the significant reliability gains.

These results suggest that MI feedback can not only improve segmentation performance in aggressive tumor regions but also produce more trustworthy uncertainty estimates for clinical decision-making.

### 4.6.6 On the Use of Predictive Entropy Over MI

A comparison between the use of Mutual Information and Predictive Entropy is also done using the BraTS 2019, LA 2018, and Pancreas-CT datasets with 10% labeled samples.

The results in Table 15 indicate a clear benefit from the use of MI over Predictive Entropy on the BraTS 2019 dataset using 10% labeled samples. Substantial gains are observed in the TC and ET regions, with increases of 1.17% and 1.61% in mean DSC, respectively. Two-sided paired t-tests confirm these improvements are

Table 16: Predictive Entropy vs. MI on LA 2018 and Pancreas-CT with 10% labeled samples

| MI / $\mathcal{H}$ | DSC (%, ↑) | Jaccard (%, ↑) | HD95 (↓) | ASD (↓) |
|---|---|---|---|---|
| LA 2018 with 8 (10%) Labeled Samples | | | | |
| $\mathcal{H}$ | 91.44 | 84.31 | 5.44 | 1.48 |
| MI | **91.77** | **84.84** | **5.13** | **1.44** |
| Pancreas-CT with 6 (10%) Labeled Samples | | | | |
| $\mathcal{H}$ | 78.05 | 64.91 | 10.58 | 1.23 |
| MI | **80.93** | **68.42** | **6.81** | **1.17** |

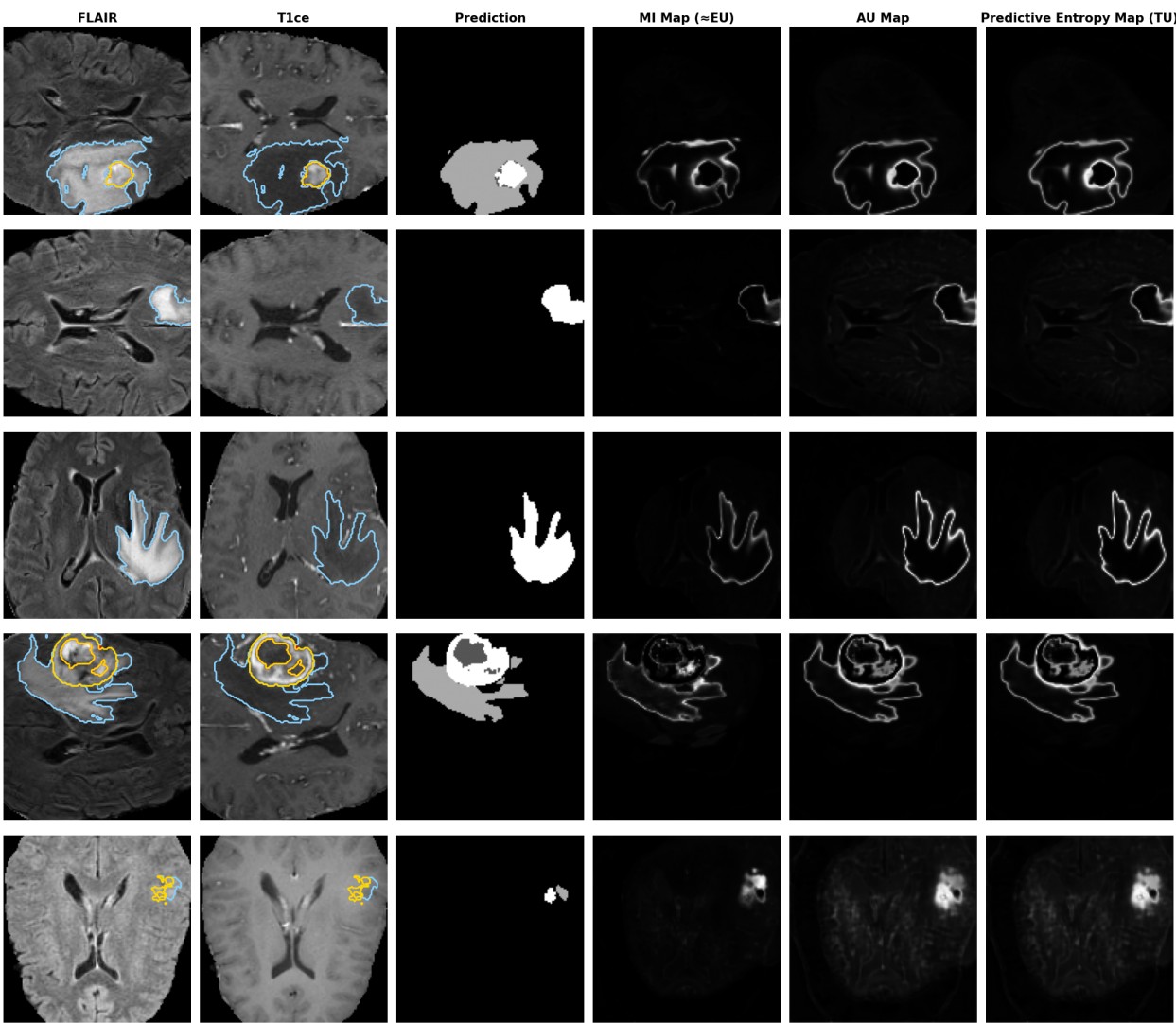

Figure 15: Visualization of the MI maps, Predictive Entropy Maps, and the computed AU map on BraTS 2019. The AU map is computed by taking the difference between the Predictive Entropy Map and the MI map (TU - EU), where TU is the total uncertainty. The contours overlaid on the FLAIR and T1ce images are derived from the ground truth masks, with blue representing PTE, red representing NCR/NET, and yellow representing ET.

statistically significant in the ET ($p < 0.01$) and WT ($p < 0.05$) regions. When analyzing Figure 14, there is a substantial decrease in the IQR when using MI over Predictive Entropy, although median DSC are comparable. From Table 16, we observe clear gains when using MI over Predictive Entropy on the LA 2018 and Pancreas-CT dataset with 10% labeled samples. The gains observed in Pancreas-CT are very significant, with a 2.88% increase in mean DSC and a decrease of 3.77 mm in mean HD95.

These observations can be explained by the fundamental difference between Predictive Entropy and MI. Predictive Entropy captures both aleatoric (AU) and epistemic (EU) uncertainty, whereas MI isolates epistemic uncertainty arising from model uncertainty. As illustrated in Figure 15, predictive entropy and MI maps have similar structures for large and homogeneous regions such as the WT. In contrast, substantial differences are observed for the TC and ET regions, as well as additional background noise due to aleatoric uncertainty. By selectively emphasizing epistemic uncertainty, MI provides a more informative feedback signal for guiding segmentation in complex and low-occupancy regions.

## 5   Discussion

Based on the experimental results and ablation studies, the proposed approach demonstrates strong performance and improved consistency compared to existing semi-supervised methods. The observed reduction in performance variability highlights the robustness of the framework, which is particularly important when handling 3D medical scans from heterogeneous patient populations. In clinical settings, such consistency across cases is often as critical as high average accuracy.

A key advantage of our approach is its computational efficiency. Unlike many state-of-the-art methods that use multiple model instances for ensembling or teacher-student frameworks, our pipeline achieves better results using a single model instance. By leveraging the stochastic nature of dropout during training to approximate uncertainty, we effectively use the model's own internal logic as a supervisory signal. This significantly reduces memory overhead during training. This is particularly useful for memory-constrained training in hospitals that don't have access to high-end computing resources.

The ablation studies highlight the importance of using Mutual Information, particularly when compared with Predictive Entropy. In this application, epistemic uncertainty alone appears to provide more effective guidance than total uncertainty, which agrees with recent findings (Pocevičiūtė et al., 2022; Cangalovic et al., 2023; Kwon et al., 2020). At the same time, these studies also emphasize the task-dependent nature of uncertainty estimation, as the relative utility of different measures has been shown to vary across applications (Wimmer et al., 2023).

However, the proposed approach does have certain limitations. While the approach can be extended to different architectures, the effectiveness of MI feedback is dependent on both the architecture and dataset being used, showing consistent benefits on concatenation-based models. Furthermore, even though the proposed approach competes with and even outperforms other methods overall, it may still be inadequate for real-world clinical applications, where extreme precision and accuracy are critical, especially in the segmentation of very small tumors or lesions.

## 6   Conclusion

This paper proposes a novel approach to perform 3D medical image segmentation with limited labeled samples. Our method leverages Mutual Information computed from Monte-Carlo Dropout operations performed on each decoder as feedback in the form of confidence gating, while using a single model instance. The results obtained from the experiments conducted show consistent improvement over existing approaches on the BraTS 2019, BraTS-GLI 2024, LA 2018, and NIH Pancreas-CT datasets. These results also show applicability across single and multiple modalities, binary and multilabel segmentation, while also showing reduced variability and training efficiency in addition to improved segmentation performance.

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
