# OpenReview forum: "MINDFeed: Mutual Information-Guided Single-Network Consistency Learning for Semi-Supervised 3D Medical Image Segmentation"
_TMLR — Under review for TMLR_

### Review · Reviewer_L7Es · 2026-05-18

**Summary Of Contributions:**

The paper proposes MINDFeed, a semi-supervised 3D medical image segmentation framework that computes decoder-level mutual information from Monte-Carlo dropout and uses the resulting uncertainty maps as feature-level feedback gates. The method is positioned as a single-network alternative to teacher-student, EMA, and multi-model consistency frameworks, while still using deep supervision, weak-strong consistency, 3D CutMix, and pseudo-labeling. Experiments are conducted on BraTS 2019, BraTS-GLI 2024, LA 2018, and NIH Pancreas-CT, covering both MRI and CT, binary and multi-region segmentation, and low-label regimes. The reported results show consistent improvements over several semi-supervised baselines, with additional ablations on deep supervision, CutMix, gating location, MI feedback, and predictive entropy versus MI.

**Audience:**

Yes

**Audience Explanation:**

/

**Claims And Evidence:**

Yes

**Claims Explanation:**

/

**Requested Changes:**

1.	The core operation is essentially a soft confidence gate based on 1 minus the estimated uncertainty map, applied to intermediate features. This is related to a broad family of uncertainty masking, confidence weighting, attention gating, and reliable-region consistency methods. The paper argues that the distinction is “feedback gating” rather than loss modification, but the conceptual leap is not sufficiently strong unless the authors more carefully distinguish it from prior uncertainty-guided feature modulation, attention masking, confidence-based pseudo-label filtering, and uncertainty-aware consistency regularization.
2.	MINDFeed combines several powerful ingredients: deep supervision, MC dropout ensemble-like prediction, weak-strong augmentation, 3D CutMix, multi-decoder pseudo-label averaging, KL consistency, and MI-based gating. The ablations show that MI helps, but the full system is not decomposed against equally strong variants such as deep supervision plus CutMix plus weak-strong consistency without multi-decoder pseudo-label averaging, confidence-thresholded pseudo-labeling, entropy-based soft weighting, or feature gating with random/smoothed/boundary-aware maps. Since the paper itself shows that deep supervision and CutMix already contribute substantially, the current evidence does not prove that the claimed mechanism is the decisive factor.
3.	For BraTS datasets, the authors state that all methods are trained under the same 3D UNet backbone and inference settings, but for LA 2018 and Pancreas-CT, several results are quoted from prior papers while others are obtained using official implementations. This mixture makes the comparison sensitive to splits, preprocessing, implementation details, training iterations, augmentation policies, and hyperparameter tuning. A strict comparison should retrain all baselines under identical splits, codebase, hardware-independent settings, and validation protocol, or at least provide exact split files, seeds, and complete implementation details.
4.	The paper reports paired t-tests over test cases, but it does not appear to report results over multiple independent training runs or different labeled/unlabeled splits. In semi-supervised medical segmentation, variance across random labeled subsets and initialization can be large, especially with 6, 8, 12, or 16 labeled volumes. A paired test over one trained model’s test cases does not establish robustness to training randomness. The paper should report mean ± standard deviation over multiple seeds and multiple labeled splits, and then test significance over runs or splits rather than only over test samples.
5.	Although the method does not maintain a separate teacher or second model, it performs T stochastic forward passes for weak inputs and uses multiple decoder heads. With T = 5 and Ndecoders = 4, the pseudo-label and MI computation uses many stochastic decoder outputs per iteration. Memory may be lower than multi-model approaches, but wall-clock training time, throughput, and energy cost are not reported. A method cannot be claimed as computationally efficient based mainly on GPU memory, especially when Table 5 even compares memory numbers obtained partly on different GPUs.

---

### Review · Reviewer_oPLv · 2026-05-25

**Summary Of Contributions:**

Authors of this paper proposed MINDFeed for 3D medical image segmentation in semi-supervised setting, which leverages mutual information computed from Monte-Carlo Dropout operations performed on each decoder as feedback in the form of confidence gating. Extensive experiments on four benchmark datasets demonstrate the effectiveness and efficiency of the proposed method.

**Audience:**

Yes

**Audience Explanation:**

This work covers some areas of interesting for TMLR’s audience including the uncertainty-aware learning based on mutual information and semi-supervised learning with few training samples. In addition, the application is specifically targeting on the 3D medical image segmentation, which is a hot research area.

**Broader Impact Concerns:**

I have no concern on the ethical implications of the work.

**Claims And Evidence:**

Yes

**Claims Explanation:**

The feedback gating mechanism was built on existing works on calculating mutual information, while the gating is a bit ad hoc. There lacks the analysis on how the modification of layer output could have expected impact on the segmentation output.

The loss function used for training consists of supervised and unsupervised components. Authors claimed that the learning paradigm is different from prior approaches like modifications to losses. However, the proposed training is also a tradeoff two terms. It is unclear what the key novelty the proposed training approach possesses.

The proposed method was extensively tested on four benchmark datasets and two model architectures. Results based on various evaluation criterions as well as visual comparisons demonstrated the effectiveness of the proposed method. Ablation study also showed the impact of each component.

**Requested Changes:**

The proposed feedback gating mechanism needs more clarifications. It is unclear that how (5) can affect the segmentation output from loss optimization perspective, even though the intuition is to rescale the activation value of certain layer. What is the base of log in (6)? If it is 2 as used in Shannon entropy, $\log(C)$ will be no less than 1 if C>=2. What is the interpolation function?  Certain derivation details from (1) to (2) and (3) can be included for better understanding the proposed method.

It might be good to specify how the weight $\lambda$ is computed in Algorithm 1 and provide sufficient context for 3D CutMix method on how $\tilde{X}_{strong}$ is obtained based on weak augmentation and MI maps.

It might be better to highlight the novelty of this paper and include sufficient context of existing works.

---

### Review · Reviewer_LYM3 · 2026-07-01

**Summary Of Contributions:**

This manuscript proposes MINDFeed, a single-network semi-supervised framework for 3D medical image segmentation. The core idea is to use Mutual Information (MI) as a voxel-level feedback gating signal to modulate intermediate features during training. The method is evaluated on four public benchmarks and achieves competitive or superior performance compared with recent semi-supervised approaches.

Strengths：
1、The shift from using uncertainty as a loss modifier to using it as a feature gating signal is conceptually interesting.
2、The method is evaluated across diverse settings, including multi-modal (BraTS) vs. single-modal (LA, Pancreas), multi-class vs. binary segmentation. The consistent improvements across these benchmarks suggest good generalizability.
3、The ablation studies are comprehensive and provide sufficient empirical support for the proposed design choices.

Weaknesses
1、The paper claims that MI gating "softly modulates decoder features" (Sec. 3.3), yet it provides no visualization of how feature representations change before and after gating.
2、The manuscript does not provide a code repository, and some implementation details are insufficient to ensure reproducibility.
3、The manuscript would benefit from careful proofreading to correct several minor grammatical and formatting inconsistencies (e.g., repeated equation references such as "equation equation 3").

**Audience:**

Yes

**Audience Explanation:**

The paper investigates a relevant problem in semi-supervised 3D medical image segmentation and presents a technically sound solution with comprehensive experimental validation. The proposed use of mutual information as a feature gating signal offers a useful perspective that may be of interest to researchers working on semi-supervised learning and uncertainty-aware deep learning.

**Broader Impact Concerns:**

No concerns.

**Claims And Evidence:**

Yes

**Claims Explanation:**

Overall, the experimental evidence is sufficient to support the main claims of the paper. The evaluation covers multiple datasets and includes comprehensive ablation studies that validate the proposed components. Although additional qualitative analysis and public code would improve the strength and reproducibility of the work, the presented evidence is generally convincing.

**Requested Changes:**

Critical：
1、Provide qualitative visualizations illustrating the effect of MI-based gating on intermediate feature representations, such as feature maps before and after gating.
2、Carefully proofread the manuscript to correct minor grammatical and formatting inconsistencies, such as repeated equation references (e.g., "equation equation 3").
Suggested：
1、Improve reproducibility by making the code publicly available.